# Exposing distinct subcortical components of the auditory brainstem response evoked by continuous naturalistic speech

Melissa J Polonenko[1,2,3], Ross K Maddox[1,2,3,4]*

[1]Department of Neuroscience, University of Rochester, Rochester, United States; [2]Del Monte Institute for Neuroscience, University of Rochester, Rochester, United States; [3]Center for Visual Science, University of Rochester, Rochester, United States; [4]Department of Biomedical Engineering, University of Rochester, Rochester, United States

**Abstract** Speech processing is built upon encoding by the auditory nerve and brainstem, yet we know very little about how these processes unfold in specific subcortical structures. These structures are deep and respond quickly, making them difficult to study during ongoing speech. Recent techniques have begun to address this problem, but yield temporally broad responses with consequently ambiguous neural origins. Here, we describe a method that pairs re-synthesized 'peaky' speech with deconvolution analysis of electroencephalography recordings. We show that in adults with normal hearing the method quickly yields robust responses whose component waves reflect activity from distinct subcortical structures spanning auditory nerve to rostral brainstem. We further demonstrate the versatility of peaky speech by simultaneously measuring bilateral and ear-specific responses across different frequency bands and discuss the important practical considerations such as talker choice. The peaky speech method holds promise as a tool for investigating speech encoding and processing, and for clinical applications.

*For correspondence:
ross.maddox@rochester.edu

Competing interests: The authors declare that no competing interests exist.

## Introduction

Understanding speech is an important, complex process that spans the auditory system from cochlea to cortex. A temporally precise network transforms the strikingly dynamic fluctuations in amplitude and spectral content of natural, ongoing speech into meaningful information and modifies that information based on attention or other priors (*Mesgarani et al., 2009*). Subcortical structures play a critical role in this process – they do not merely relay information from the periphery to the cortex, but also perform important functions for speech understanding, such as localizing sound (e.g., *Grothe and Pecka, 2014*) and encoding vowels across different levels and in background noise (e.g., *Carney et al., 2015*). Furthermore, subcortical structures receive descending information from the cortex through corticofugal pathways (*Bajo et al., 2010*; *Bajo and King, 2012*; *Winer, 2005*), suggesting that they may also play an important role in modulating speech and auditory streaming. Given the complexity of speech processing, it is important to parse and understand contributions from different neural generators. However, these subcortical structures are deep and respond to stimuli with very short latencies, making them difficult to study during ecologically salient stimuli such as continuous and naturalistic speech. We created a novel paradigm aimed at elucidating the contributions from distinct subcortical structures to ongoing, naturalistic speech.

Activity in deep brainstem structures can be 'imaged' by the latency of waves in a surface electrical potential (electroencephalography [EEG]) called the auditory brainstem response (ABR). The ABR's component waves have been attributed to activity in different subcortical structures with characteristic latencies: the auditory nerve contributes to waves I and II (~1.5–3 ms), the cochlear nucleus

to wave III (~4 ms), the superior olivary complex and lateral lemniscus to wave IV (~5 ms), and the lateral lemniscus and inferior colliculus to wave V (~6 ms) (*Møller and Jannetta, 1983*; review by *Moore, 1987*; *Starr and Hamilton, 1976*). Waves I, III, and V are most often easily distinguished in the human response. Subcortical structures may also contribute to the earlier $P_0$ (12–14 ms) and $N_a$ (15–25 ms) waves (*Hashimoto, 1982*; *Kileny et al., 1987*; *Picton et al., 1974*) of the middle latency response (MLR), which are then followed by thalamo-cortically generated waves $P_a$, $N_b$, and $P_b/P_1$ (*Geisler et al., 1958*; *Goldstein and Rodman, 1967*). ABR and MLR waves have a low signal-to-noise ratio (SNR) and require numerous stimulus repetitions to record a good response. Furthermore, they are quick and often occur before the stimulus has ended. Therefore, out of necessity, most human brainstem studies have focused on brief stimuli such as clicks, tone pips, or speech syllables rather than more natural speech.

Recent analytical techniques have overcome limitations on stimuli, allowing continuous naturally uttered speech to be used. One such technique extracts the fundamental waveform from the speech stimulus and finds the envelope of the cross-correlation between that waveform and the recorded EEG data (*Forte et al., 2017*). The response has an average peak time of about 9 ms, with contributions primarily from the inferior colliculus (*Saiz-Alía and Reichenbach, 2020*). A second technique considers the rectified broadband speech waveform as the input to a linear system and the EEG data as the output, and uses deconvolution to compute the ABR waveform as the impulse response of the system (*Maddox and Lee, 2018*). The speech-derived ABR shows a wave V peak whose latency is highly correlated with the click response wave V across subjects, demonstrating that the component is generated in the rostral brainstem. A third technique averages responses to each chirp (click-like transients that quickly increase in frequency) in re-synthesized 'cheech' stimuli (CHirp spEECH; *Backer et al., 2019*) that interleaves alternating octave frequency bands of speech and chirps aligned with some glottal pulses. Brainstem responses to these stimuli also show a wave V, but do not show earlier waves (*Backer et al., 2019*). While these methods reflect subcortical activity, the first two provide temporally broad responses with a lack of specificity regarding underlying neural sources. None of the three methods shows the earlier canonical components such as waves I and III that would allow rostral brainstem activity to be distinguished from, for example, the auditory nerve. Such activity is important to assess, especially given the current interest in the potential contributions of auditory nerve loss in disordered processing of speech in noise (*Bramhall et al., 2019*; *Liberman et al., 2016*; *Prendergast et al., 2017*).

Although click responses can assess sound encoding (of clicks) at early stages of the auditory system, a speech-evoked response with the same components would assess subcortical structure-specific encoding within the acoustical context of the dynamic spectrotemporal characteristics of speech – information that is not possible to obtain from click responses. Furthermore, changes to the amplitudes and latencies of these early components could inform our understanding of speech processing if deployed in experiments that compare conditions requiring different states of processing, such as attended/unattended speech or understood/foreign language. For example, if a wave I from the auditory nerve differed between speech stimuli that were attended versus unattended, then this would add to our current understanding of the brainstem's role in speech processing. Therefore, a click-like response that is evoked by speech stimuli facilitates new investigations into speech encoding and processing.

Thus, we asked if we could further assess underlying speech encoding and processing in multiple distinct early stages of the auditory system by (1) evoking additional waves than wave V of the canonical ABR and (2) measuring responses to different frequency ranges of speech (corresponding to different places of origin on the cochlea). The ABR is strongest to very short stimuli such as clicks, so we created 'peaky' speech. The design goal of peaky speech is to re-synthesize natural speech so that its defining spectrotemporal content is unaltered – maintaining the speech as intelligible and identifiable – but its pressure waveform consists of maximally sharp peaks so that it drives the ABR as effectively as possible (giving a very slight 'buzzy' quality when listening under good headphones; *Audio files 1–6*). The results show that peaky speech evokes canonical brainstem responses and frequency-specific responses, paving the way for novel studies of subcortical contributions to speech processing.

## Results

### Broadband peaky speech yields more robust responses than unaltered speech

Broadband peaky speech elicits canonical brainstem responses

We re-synthesized speech to be 'peaky' with the primary aim to evoke additional, earlier waves of the ABR that identify different neural generators. Indeed, *Figure 1* shows that waves I, III, and V of the canonical ABR are clearly visible in the group average and the individual responses to broadband peaky speech, which were filtered at a typical high-pass cutoff of 150 Hz to highlight the earlier ABR waves. This means that broadband peaky speech, unlike the unaltered speech, can be used to assess naturalistic speech processing at discrete parts of the subcortical auditory system, from the auditory nerve to rostral brainstem.

Waves I and V were identifiable in responses from all subjects ($N$ = 22), and wave III was identifiable in 19 of the 22 subjects. The numbers of subjects with identifiable waves I and III in these peaky speech responses were similar to the 24 and 16 out of 24 subjects for the click-evoked responses in *Maddox and Lee, 2018*. These waves are marked on the individual responses in *Figure 1*. Mean ± SEM peak latencies for ABR waves I, III, and V were 3.23 ± 0.09 ms, 5.51 ± 0.07 ms, and 7.22 ± 0.07 ms, respectively. These mean peak latencies are shown superimposed on the group average response in *Figure 1* (bottom right). Inter-wave latencies were 2.24 ± 0.06 ms ($N$ = 19) for I–III, 1.68 ± 0.05 ms ($N$ = 19) for III–V, and 4.00 ± 0.08 ($N$ = 22) for I–V. These peak inter-wave latencies fall within a range expected for brainstem responses, but the absolute peak latencies were later than those reported for a click ABR at a level of 60 dB sensation level (SL) and rate between 50 and 100 Hz (*Burkard and Hecox, 1983*; *Chiappa et al., 1979*; *Don et al., 1977*).

### More components of the ABR and MLR are present with broadband peaky than unaltered speech

Having established that broadband peaky speech evokes robust canonical ABRs, we next compared both ABR and MLR responses to those evoked by unaltered speech. To simultaneously evaluate ABR and MLR components, a high-pass filter with a 30-Hz cutoff was used (unlike the broadband peaky response, the 150 Hz high-pass cutoff does not reveal earlier components in the response to unaltered speech). *Figure 2A* shows that overall there were morphological similarities between responses to both types of speech; however, there were more early and late component waves to broadband peaky speech. More specifically, whereas both types of speech evoked waves V, $N_a$, and $P_a$, broadband peaky speech also evoked waves I, often III (14–19 of 22 subjects depending on whether a 30 or 150 Hz high-pass filter cutoff was used), and $P_0$. With a lower cutoff for the high-pass filter, wave III rode on the slope of wave V and was less identifiable in the grand average shown in *Figure 2A* than that shown with a higher cutoff in *Figure 1*. Wave V was more robust and sharper to broadband peaky speech but peaked slightly later than the broader wave V to unaltered speech. For reasons unknown to us, the half-rectified speech method missed the MLR wave $P_0$, and consequently had a broader and earlier $N_a$ than the broadband peaky speech method, though this missing $P_0$ was consistent with the results of *Maddox and Lee, 2018*. These waveforms indicate that broadband peaky speech is better than unaltered speech at evoking canonical responses that distinguish activity from distinct subcortical and cortical neural generators.

Peak latencies for the waves common to both types of speech are shown in *Figure 2B*. As suggested by the waveforms in *Figure 2A*, mean ± SEM peak latencies for waves V, $N_a$, and $P_a$ were longer for broadband peaky than unaltered speech by 0.87 ± 0.06 ms (independent *t*-test, $t_{(21)}$ = 14.7, p<0.01, $d$ = 3.22), 6.92 ± 0.44 ms ($t_{(21)}$ = 15.4, p<0.01, $d$ = 3.44), and 1.07 ± 0.45 ms ($t_{(20)}$ = 2.3, p=0.032, $d$ = 0.52), respectively.

The response to broadband peaky speech showed a small but consistent response at negative and early positive lags (i.e., pre-stimulus) when using the pulse train as a regressor in the deconvolution, particularly when using the lower high-pass filter cutoff of 30 Hz (*Figure 2A*) compared to 150 Hz (*Figure 1*). For a perfectly linear causal system, this would not be expected. To better understand the source of this pre-stimulus component – and to determine whether later components were influencing the earliest components – we performed two simulations. (1) A simple linear deconvolution model in which EEG for each 64 s epoch was simulated by convolving the rectified broadband peaky

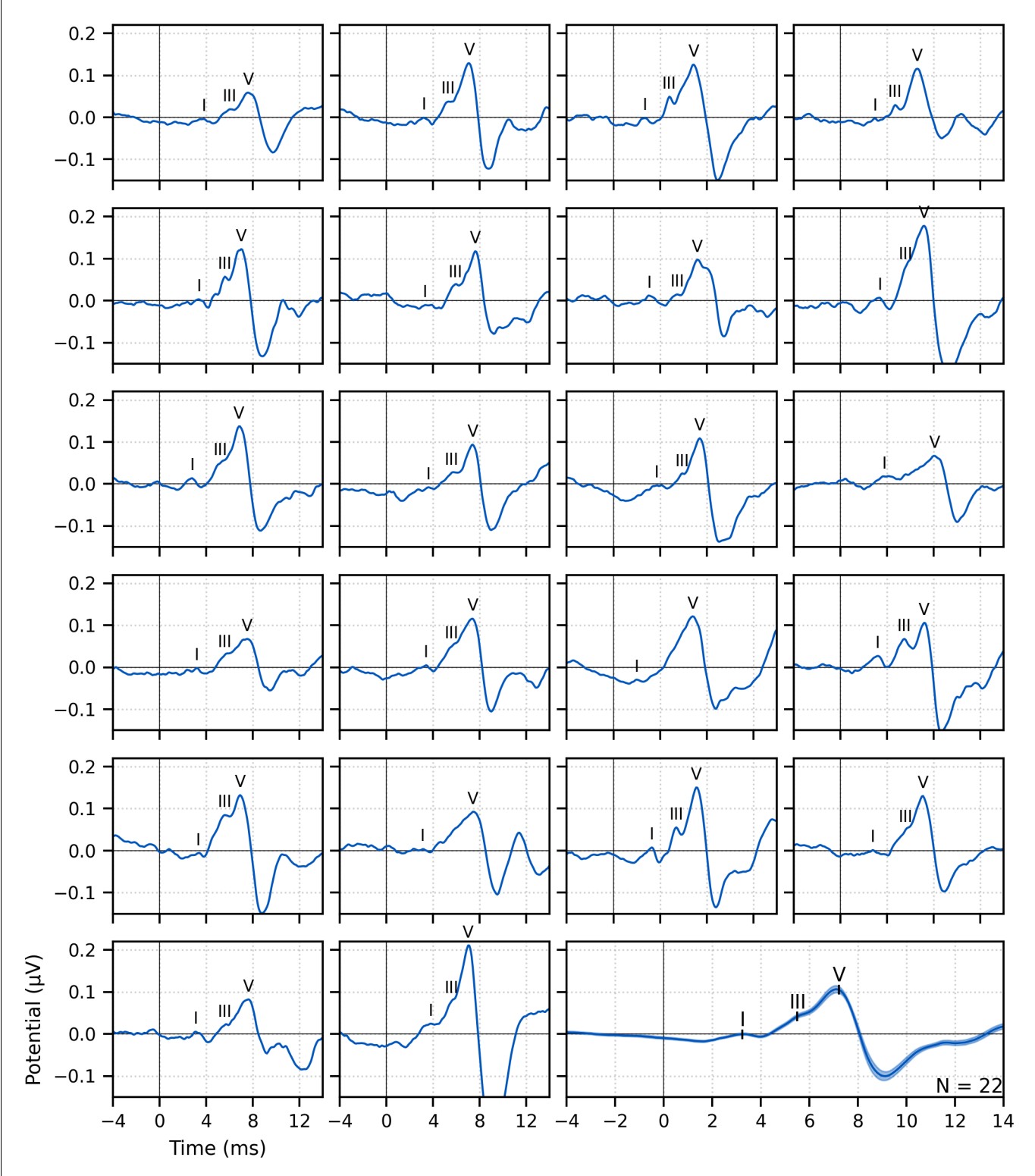

**Figure 1.** Single-subject and group average (bottom right) weighted-average auditory brainstem responses (ABRs) to ~43 min of broadband peaky speech. Areas for the group average show ±1 SEM. Responses were high-pass filtered at 150 Hz using a first-order Butterworth filter. Waves I, III, and V of the canonical ABR are evident in most of the single-subject responses (N = 22, 16, and 22, respectively) and are marked by the average peak latencies on the average response.

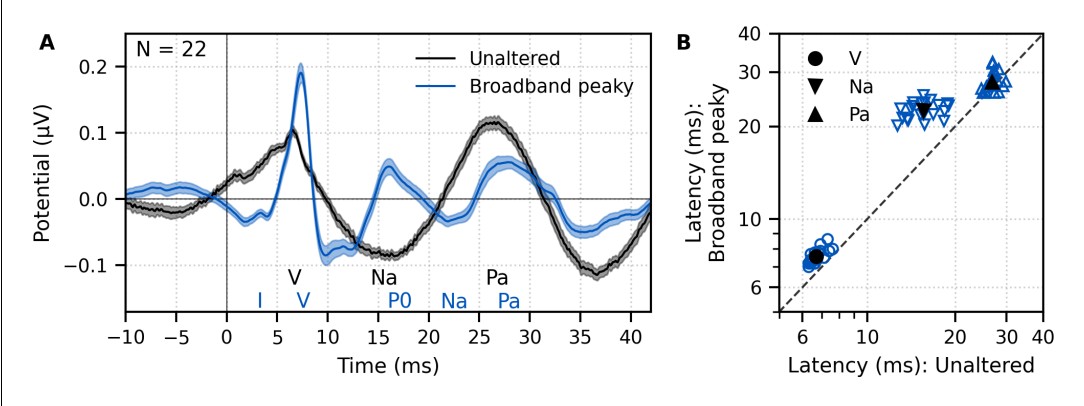

**Figure 2.** Comparison of auditory brainstem response (ABR) and middle latency response (MLR) to ~43 min each of unaltered speech and broadband peaky speech. (**A**) The average waveform to broadband peaky speech (blue) shows additional, and sharper, waves of the canonical ABR and MLR than the broader average waveform to unaltered speech (black). Responses were high-pass filtered at 30 Hz with a first-order Butterworth filter. Areas show ±1 SEM. (**B**) Comparison of peak latencies for ABR wave V (circles) and MLR waves $N_a$ (downward triangles) and $P_a$ (upward triangles) that were common between responses to broadband peaky and unaltered speech. Blue symbols depict individual subjects, and black symbols depict the mean.

speech audio with an ABR kernel that did not show the pre-stimulus component (*Figure 3—figure supplement 1*): the average broadband peaky speech ABR from 0 to 16 ms was zero-padded from 0 ms to the beginning of wave I (1.6 ms), windowed with a Hann function, normalized, and then zero-padded from −16 ms to 0 ms to center the kernel. (2) A more nuanced and well-known model of the auditory nerve and periphery that accounts for acoustics and some of the nonlinearities of the auditory system (*Rudnicki et al., 2015*; *Verhulst et al., 2018*; *Zilany et al., 2014*) – we simulated EEG for waves I, III, and V using the framework described by *Verhulst et al., 2018*, but due to computation constraints, modified the implementation to use the peripheral model by *Zilany et al., 2014*. The simulated EEG of each model was then deconvolved with the pulse trains to derive the modeled responses, which are shown in comparison with the measured response in *Figure 3*. The pre-stimulus component of the measured response was present in both models, suggesting that there are nonlinear parts of the response that are not accounted for in the deconvolution with the pulse train regressor. However, the pre-stimulus component was temporally broad compared to the components representing activity from the auditory nerve and cochlear nucleus (i.e., waves I and III), and could thus be dealt with by high-pass filtering. The pre-stimulus component was reduced with a 150-Hz first-order Butterworth high-pass filter (*Figure 1*) and minimized with a more aggressive 200-Hz second-order Butterworth high-pass filter (*Figure 3*, bottom). As expected from more aggressive high-pass filtering, wave V became smaller, sharper, and earlier, and was followed by a significant negative deflection. Therefore, when doing an experiment where the analysis needs to evaluate specific contributions to the earliest ABR components, we recommend high-pass filtering to help mitigate the complex and time-varying nonlinearities inherent in the auditory system, as well as potential influences by responses from later sources.

We verified that the EEG data collected in response to broadband peaky speech could be regressed with the half-wave rectified speech to generate a response. *Figure 4A* shows that the derived responses to unaltered and broadband peaky speech were similar in morphology when using the half-wave rectified audio as the regressor. Correlation coefficients from the 22 subjects for the 0–40 ms lags had a median (interquartile range) of 0.84 (0.72–0.91). This means that the same EEG collected to broadband peaky speech can be flexibly used to generate the robust canonical brainstem response to the pulse train, as well as the broader response to the half-wave rectified speech.

Although responses to broadband peaky speech can be derived from both the half-wave rectified audio and pulse train regressors, the same cannot be said about unaltered speech. We also regressed the EEG data collected in response to unaltered speech with the pulse train. *Figure 4B* shows that simply using the pulse train as a regressor does not give a robust canonical response – the response contained a wave that slightly resembled wave V of the response to broadband peaky

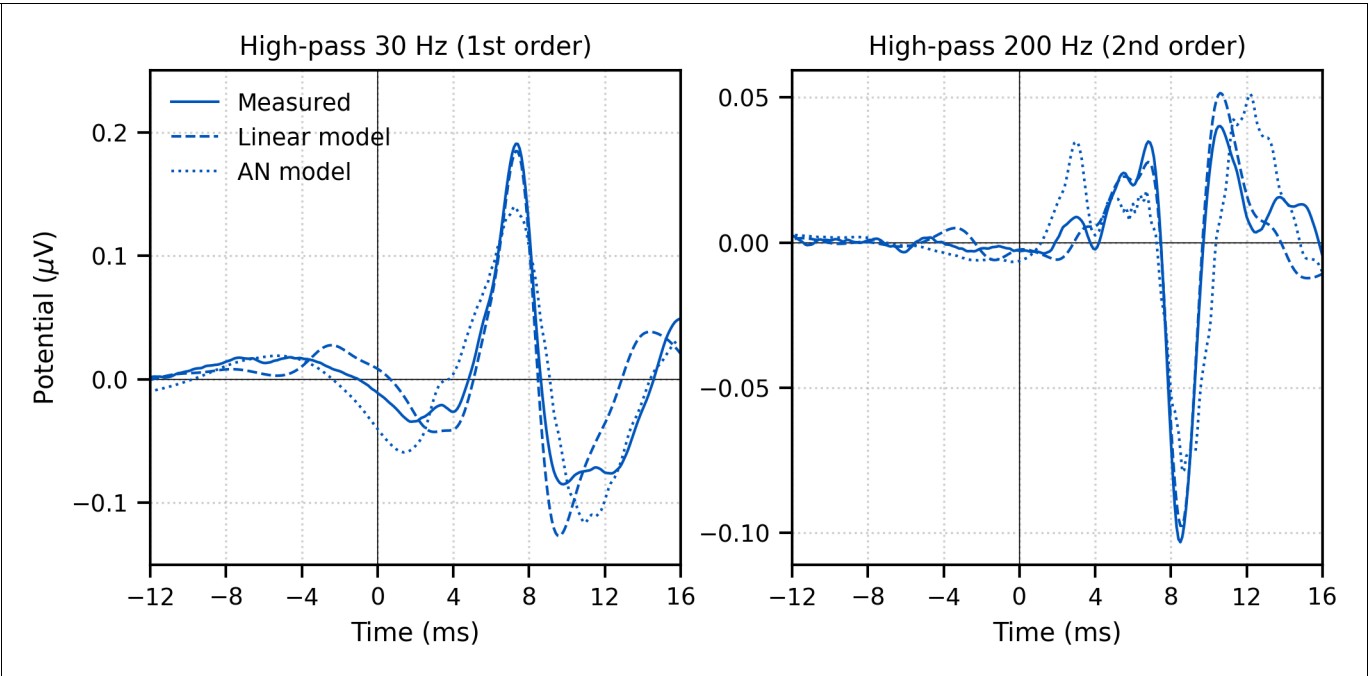

**Figure 3.** Comparison of grand average (N = 22) measured and modeled responses to ~43 min of broadband peaky speech. Amplitudes of the linear (dashed line) and auditory nerve (AN; dotted line) modeled responses were in arbitrary units, and thus scaled to match the amplitude of the measured response (solid line) over the 0–20 ms lags. The pre-stimulus component was present in all three responses using a first-order 30 Hz high-pass Butterworth filter (left column), but was minimized by aggressive high-pass filtering with a second-order 200 Hz high-pass Butterworth filter (right column).

The online version of this article includes the following figure supplement(s) for figure 3:

**Figure supplement 1.** Auditory brainstem response (ABR) kernel used for the simple linear deconvolution model.

speech, albeit at a later latency and smaller amplitude, but there were no other earlier waves of the ABR or later waves of the MLR. The correlation coefficients comparing the unaltered and broadband peaky speech responses to the pulse train had a median (interquartile range) of 0.20 (0.05–0.39). The response morphology was abnormal, with an acausal response at 0 ms and a smearing of the response in time, particularly at lags of the MLR – these effects are consistent with some phases in the unaltered speech that come pre-pulse, which differs from that of the peaky speech that has aligned phases at each glottal pulse. The aligned phases of the broadband peaky response allow for the distinct waves of the canonical brainstem and MLRs to be derived using the pulse train regressor.

## Broadband peaky speech responses differ across talkers

We next sought to determine whether response morphologies depended on the talker identity. To determine to what extent the morphology and robustness of peaky speech responses depend on a specific narrator's voice and fundamental frequency – and therefore, rate of stimulation – we compared waveforms and peak wave latencies of male- and female-narrated broadband peaky speech in 11 subjects.

The group average waveforms to female- and male-narrated broadband peaky speech showed similar canonical morphologies but were smaller and later for female-narrated ABR responses (*Figure 5A*), much as they would be for click stimuli presented at higher rates (e.g., *Burkard et al., 1990*; *Burkard and Hecox, 1983*; *Chiappa et al., 1979*; *Don et al., 1977*; *Jiang et al., 2009*). All component waves of the ABR and MLR were visible in the group average, although fewer subjects exhibited a clear wave III in the female-narrated response (9 versus all 11 subjects). The median (interquartile range) male–female correlation coefficients were 0.68 (0.56–0.78) for ABR lags of 0–15 ms with a 150 Hz high-pass filter and 0.53 (0.50–0.60) for ABR/MLR lags of 0–40 ms with a 30 Hz

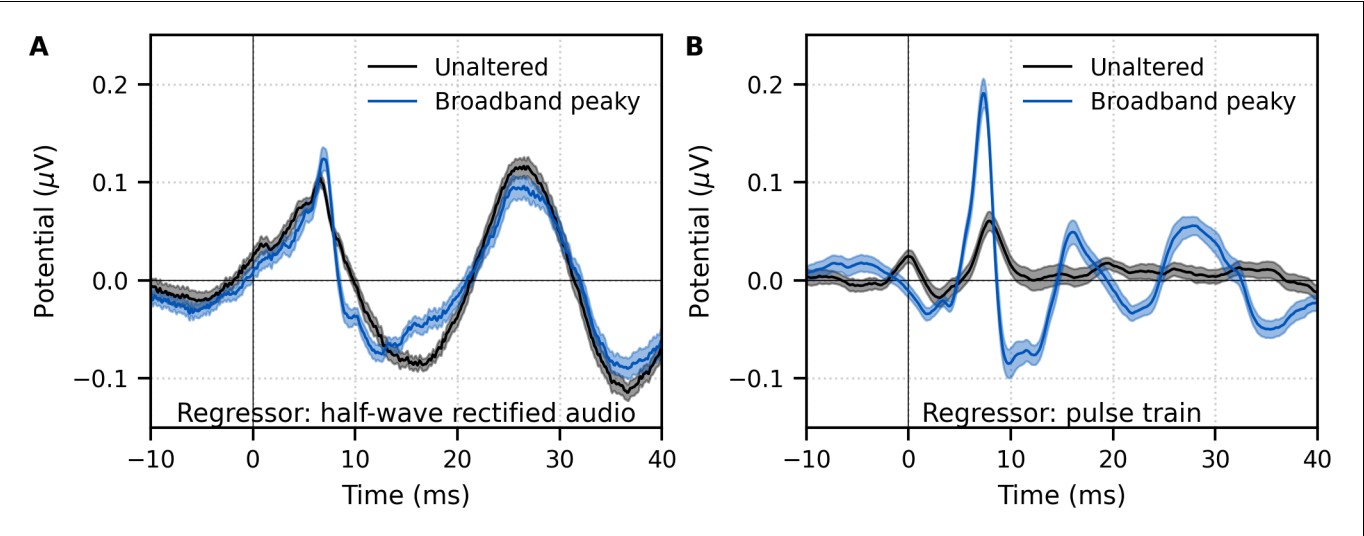

**Figure 4.** Comparison of responses derived by using the same type of regressor in the deconvolution. Average waveforms (areas show ±1 SEM) are shown for ~43 min each of unaltered speech (black) and broadband peaky speech (blue). EEG was regressed with the (**A**) half-wave rectified audio and (**B**) pulse train. Responses were high-pass filtered at 30 Hz using a first-order Butterworth filter.

The online version of this article includes the following figure supplement(s) for figure 4:

**Figure supplement 1.** Comparison of responses derived by using the half-wave rectified audio as the regressor in the deconvolution with electroencephalography (EEG) recorded in response to ~43 min of unaltered speech and multiband peaky speech.

high-pass filter (*Figure 5B*). This stimulus dependence was significantly different than the variability introduced by averaging only half the epochs (i.e., splitting by male- and female-narrated epochs) – the correlation coefficients for the data reanalyzed into even and odd epochs (each of the even/odd splits contained the same number of male- and female-narrated epochs) had a median (interquartile range) of 0.89 (0.79–0.95) for ABR lags and 0.66 (0.39–0.78) for ABR/MLR lags. These odd–even coefficients were significantly higher than the male–female coefficients for the ABR ($W_{(10)}$ = 0.0, p=0.001; Wilcoxon signed-rank test) but not when the response included all lags of the MLR ($W_{(10)}$ = 17.0, p=0.175), which is indicative of the increased variability of these later waves. These high odd–even correlations also show that responses from a single narrator are similar even if the text spoken is different. The overall male–female narrator differences for the ABR indicate that the choice of narrator for using peaky speech impacts the morphology of the early response.

As expected from the waveforms, peak latencies of component waves differed between male- and female-narrated broadband peaky speech (*Figure 5C*). Mean ± SEM peak latency differences (female – male) for waves I, III, and V of the ABR were 0.21 ± 0.13 ms ($t_{(10)}$ = 1.59, p=0.144, $d$ = 0.50), 0.42 ± 0.11 ms ($t_{(9)}$ = 3.47, p=0.007, $d$ = 1.16), and 0.54 ± 0.09 ms ($t_{(10)}$ = 5.56, p<0.001, $d$ = 1.76), respectively. Latency differences were not significant for MLR peaks ($P_0$: −0.89 ± 0.40 ms, $t_{(9)}$ = −2.13, p=0.062, $d$ = −0.71; $N_a$: −0.91 ± 0.55 ms, $t_{(8)}$ = −1.56, p=0.158, $d$ = −0.55; $P_a$: 0.09 ± 0.55 ms, $t_{(9)}$ = −0.16, p=0.880, $d$ = −0.05).

## Multiband peaky speech yields frequency-specific brainstem responses to speech

### Frequency-specific responses show frequency-specific lags

Broadband peaky speech gives new insights into subcortical processing of naturalistic speech. Not only are brainstem responses used to evaluate processing at different stages of auditory processing but ABRs can also be used to assess hearing function across different frequencies. Traditionally, frequency-specific ABRs are measured using clicks with high-pass masking noise or frequency-specific tone pips. We tested the flexibility of using our new peaky speech technique to investigate how speech processing differs across frequency regions, such as 0–1, 1–2, 2–4, and 4–8 kHz frequency bands (for details, see 'Multiband peaky speech' section in 'Materials and methods').

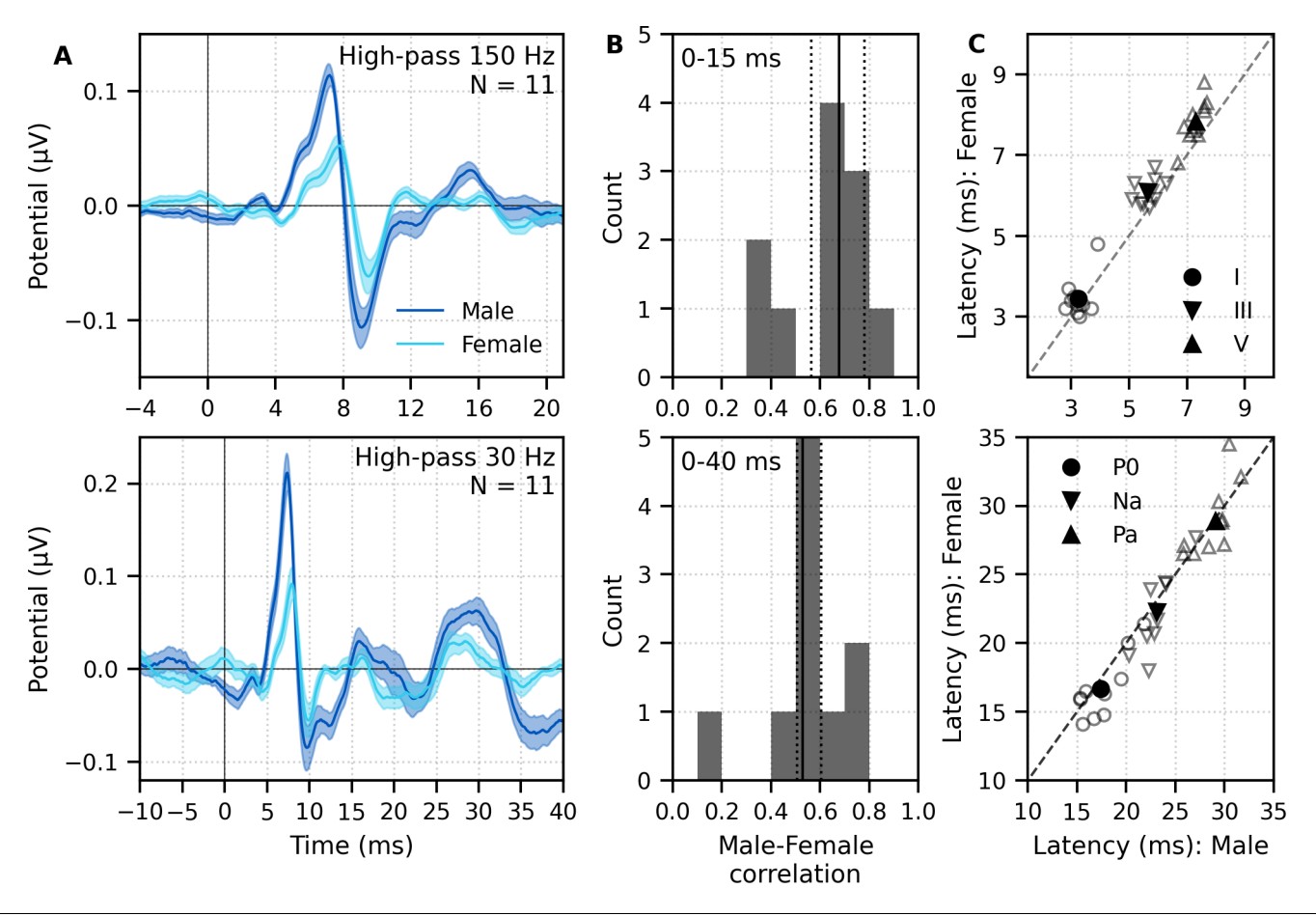

**Figure 5.** Comparison of responses to 32 min each of male- (dark blue) and female-narrated (light blue) re-synthesized broadband peaky speech. (**A**) Average waveforms across subjects (areas show ±1 SEM) are shown for auditory brainstem response (ABR) time lags with high-pass filtering at 150 Hz (top), and both ABR and middle latency response (MLR) time lags with a lower high-pass filtering cutoff of 30 Hz (bottom). (**B**) Histograms of the correlation coefficients between responses evoked by male- and female-narrated broadband peaky speech during ABR (top) and ABR/MLR (bottom) time lags. Solid lines denote the median and dotted lines the interquartile range. (**C**) Comparison of ABR (top) and MLR (bottom) wave peak latencies for individual subjects (gray) and the group mean (black). ABR and MLR responses were similar to both types of input but are smaller for female-narrated speech, which has a higher glottal pulse rate. Peak latencies for female-evoked speech were delayed during ABR time lags but faster for early MLR time lags.

Mean ± SEM responses from 22 subjects to the four frequency bands (0–1, 1–2, 2–4, and 4–8 kHz) of male-narrated multiband peaky speech are shown as colored waveforms with solid lines in *Figure 6A*. A high-pass filter with a cutoff of 30 Hz was used. Each frequency band response comprises a frequency-band-specific component as well as a band-independent common component, both of which are due to spectral characteristics of the stimuli and neural activity. The pulse trains are independent over time in the vocal frequency range – thereby allowing us to pull out responses to each different pulse train and frequency band from the same EEG – but they became coherent at frequencies lower than 72 Hz for the male-narrated speech and 126 Hz for the female speech (see Figure 15 in Materials and methods). This coherence was due to all pulse trains beginning and ending together at the onset and offset of voiced segments and was the source of the low-frequency common component of each band's response. The way to remove the common component is to calculate the common activity across the frequency band responses and subtract this waveform from each of the frequency band responses (see 'Response derivation' section in 'Materials and methods'). This common component waveform is shown by the dot-dashed gray line, which is superimposed with each response to the frequency bands in *Figure 6A*. The subtracted, frequency-specific waveforms to each frequency band are shown by the solid lines in *Figure 6B*. Of course, the

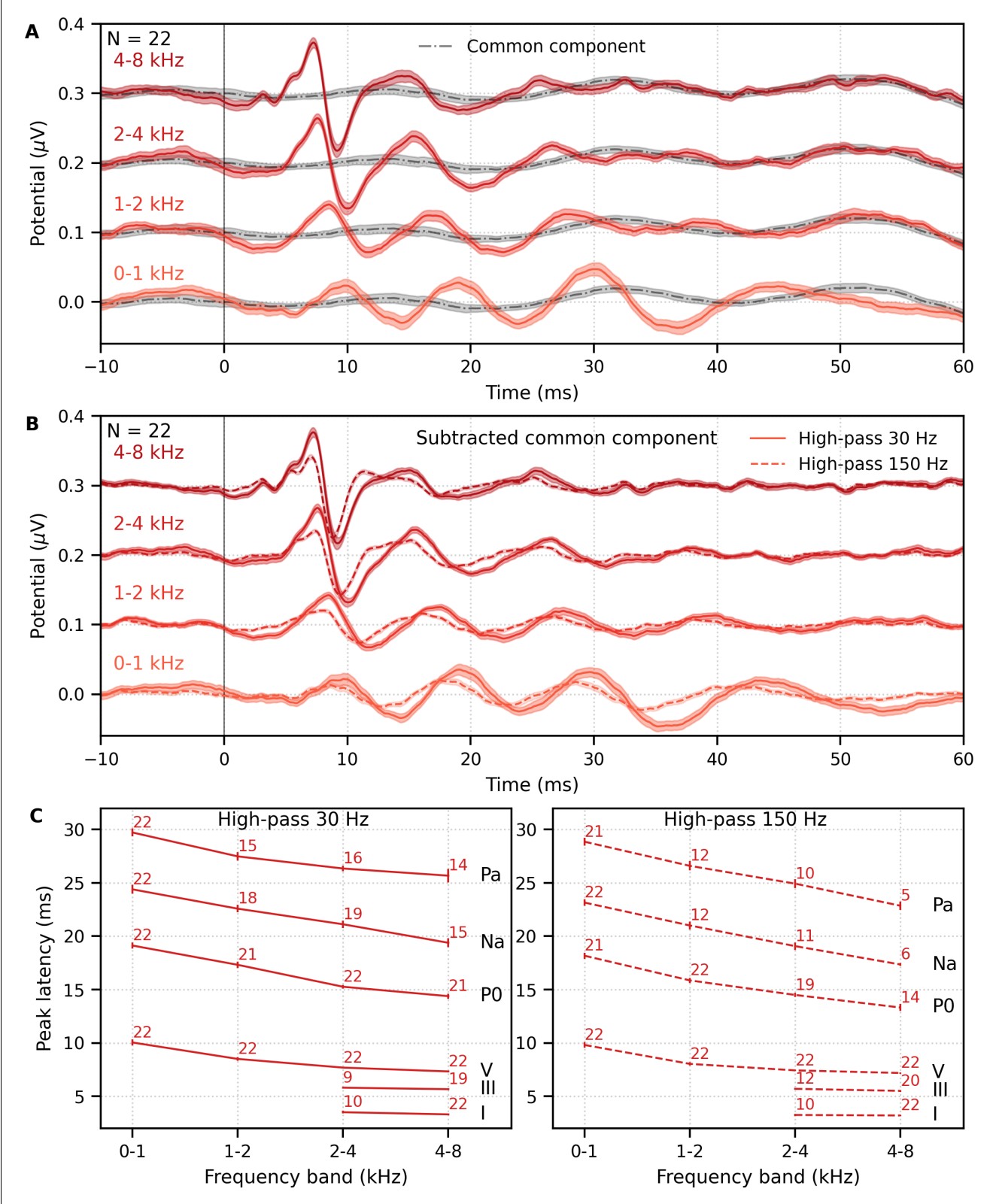

**Figure 6.** Comparison of responses to ~43 min of male-narrated multiband peaky speech. (**A**) Average waveforms across subjects (areas show ±1 SEM) are shown for each band (colored solid lines) and common component (dot-dash gray line, same waveform replicated as a reference for each band), which was calculated using six false pulse trains. (**B**) The common component was subtracted from each band's response to give the frequency-specific waveforms (areas show ±1 SEM), which are shown with high-pass filtering at 30 Hz (solid lines) and 150 Hz (dashed lines). (**C**) Mean ± SEM peak latencies

*Figure 6 continued on next page*

*Figure 6 continued*

for each wave decreased with increasing band frequency. Numbers of subjects with an identifiable wave are given for each wave and band. Details of the mixed effects models for (C) are provided in *Supplementary file 1A*.

The online version of this article includes the following figure supplement(s) for figure 6:

**Figure supplement 1.** Comparison of responses to 64 min each of male- (left) and female-narrated (right) multiband peaky speech created with the dynamic random frequency shift method.

subtracted waveforms could also then be high-pass filtered at 150 Hz to highlight earlier waves of the brainstem responses, as shown by the dashed lines in *Figure 6B*. However, this method reduces the amplitude of the responses, which in turn affects response SNR and detectability. In some scenarios, due to the low-pass nature of the common component, high-passing the waveforms at a high enough frequency may obviate the need to formally subtract the common component. For example, at least for the narrators used in these experiments, the common component contained minimal energy above 150 Hz, so if the waveforms are already high-passed at 150 Hz to focus on the early waves of the ABR, then the step of formally subtracting the common component may not be necessary. But beyond the computational cost, there is no reason not to subtract the common component and doing so allows lower filter cutoffs to be used.

Overall, the frequency-specific responses showed characteristic ABR and MLR waves with longer latencies for lower frequency bands, as would be expected from responses arising from different cochlear regions. Also, waves I and III of the ABR were visible in the group average waveforms of the 2–4 kHz (≥41% of subjects) and 4–8 kHz (≥86% of subjects) bands, whereas the MLR waves were more prominent in the 0–1 kHz (≥95% of subjects) and 1–2 kHz (≥54% of subjects) bands.

These frequency-dependent latency changes for the frequency-specific responses are highlighted further in *Figure 6C*, which shows mean ± SEM peak latencies and the number of subjects who had a clearly identifiable wave. The change in peak latency with frequency band was modeled using a power law regression (*Harte et al., 2009*; *Neely et al., 1988*; *Rasetshwane et al., 2013*; *Strelcyk et al., 2009*). The fixed effect of wave and its interaction with frequency were also included. Details of the statistical model are described in *Supplementary file 1A*. Modeled parameters that can be meaningfully compared to existing norms are given in *Table 1A*. In general, there was good agreement with these norms. There was a significant decrease in wave V latency with increasing frequency band (slope p<0.001 for 30 and 150 Hz), which was shallower (i.e., less negative) for MLR

**Table 1.** Parameter estimates and SEM for power law fits to the multiband peaky speech auditory brainstem response wave V data in the three experiments*.

| Type | Narrator | High-pass cutoff | a | b | d |
|---|---|---|---|---|---|
| Norms[†] | | | 4.70–5.00 ms | 3.46–5.39 ms | 0.22–0.50 |
| *A. Experiment 1* | | | | | |
| Diotic four-bands | Male | 30 Hz | 5.13 ± 0.08 ms | 3.95 ± 1.03 ms | 0.41 ± 0.02 |
| | | 150 Hz | 4.80 ± 0.08 ms | 3.95 ± 1.03 ms | 0.37 ± 0.02 |
| *B. Experiment 2* | | | | | |
| Diotic four-bands | Male | 30 Hz | 5.06 ± 0.14 ms | 4.42 ± 1.04 ms | 0.45 ± 0.03 |
| | Female | 30 Hz | 5.58 ± 0.12 ms | 3.94 ± 1.08 ms | 0.44 ± 0.05 |
| *C. Experiment 3* | | | | | |
| Dichotic five-bands | Male | 30 Hz | 5.06 ± 0.14 ms[‡] | 4.13 ± 1.04 ms[§] | 0.36 ± 0.02 |
| | Female | 30 Hz | 5.58 ± 0.12 ms[‡] | 3.75 ± 1.07 ms[§] | 0.41 ± 0.03 |

*Power model: $\tau(f) = a + bf^{-d}$ where $a = \tau_{synaptic} + \tau_{I-V}$ and $\tau_{synaptic} = 0.8$ ms. See 'Statistical analyses' section in 'Materials and methods' for more detail.

[†]Norms for tone pips and derived bands were calculated for 65 dBppeSPL using the model's level-dependent parameter when appropriate (*Neely et al., 1988*; *Rasetshwane et al., 2013*; *Strelcyk et al., 2009*).

[‡]Estimates from experiment 2 were used.

[§]Estimates given for the left ear; there was not a significant difference for the right ear.

waves compared to the ABR wave V (all p<0.001 for interactions between wave and frequency term). Unsurprisingly there were significantly different latencies for each MLR wave $P_0$, $N_a$, and $P_a$ compared to the ABR wave V (all effects of wave on the intercept p<0.001).

Next, the frequency-specific responses (i.e., multiband responses with common component subtracted) were summed and the common component added to derive the entire response to multiband peaky speech. As shown in *Figure 7*, this summed multiband response was strikingly similar in morphology to the broadband peaky speech. Both responses were high-passed filtered at 150 Hz and 30 Hz to highlight the earlier ABR waves and later MLR waves, respectively. The median (interquartile range) correlation coefficients from the 22 subjects were 0.90 (0.84–0.94) for 0–15 ms ABR lags and 0.66 (0.55–0.83) for 0–40 ms MLR lags. The similarity verifies that the frequency-dependent responses are complementary to each other to the common component such that these components add linearly into a 'whole' broadband response. If there were significant overlap in the cochlear regions, for example, the summed response would not resemble the broadband response to such a degree and would instead be larger. The similarity also verified that the additional changes we made to create re-synthesized multiband peaky speech did not significantly affect responses compared to broadband peaky speech.

We also verified that the EEG data collected in response to multiband peaky speech could be regressed with the half-wave rectified speech to generate a response. *Figure 4—figure supplement 1* shows that the derived responses to unaltered and multiband peaky speech were similar in morphology when using the half-wave audio as the regressor, although the multiband peaky speech response was broader in the earlier latencies. Correlation coefficients from the 22 subjects for the 0–40 ms lags had a median (interquartile range) of 0.83 (0.77–0.88). This means that the same EEG collected to multiband peaky speech can be flexibly used to generate the frequency-specific brainstem responses to the pulse train, as well as the broader response to the half-wave rectified speech.

## Frequency-specific responses also differ by narrator

We also investigated the effects of male- versus female-narrated multiband peaky speech in the same 11 subjects. As with broadband peaky speech, responses to both narrators showed similar morphology, but the responses were smaller and the MLR waves more variable for the female than the male narrator (*Figure 8A*). *Figure 8B* shows the male–female correlation coefficients for responses between 0–40 ms with a high-pass filter of 30 Hz and between 0–15 ms with a high-pass filter of 150 Hz. The median (interquartile range) male–female correlation coefficients were better for higher frequency bands, ranging from 0.12 (−0.11–0.34) for the 1–2 kHz band to 0.44 (0.32–0.49)

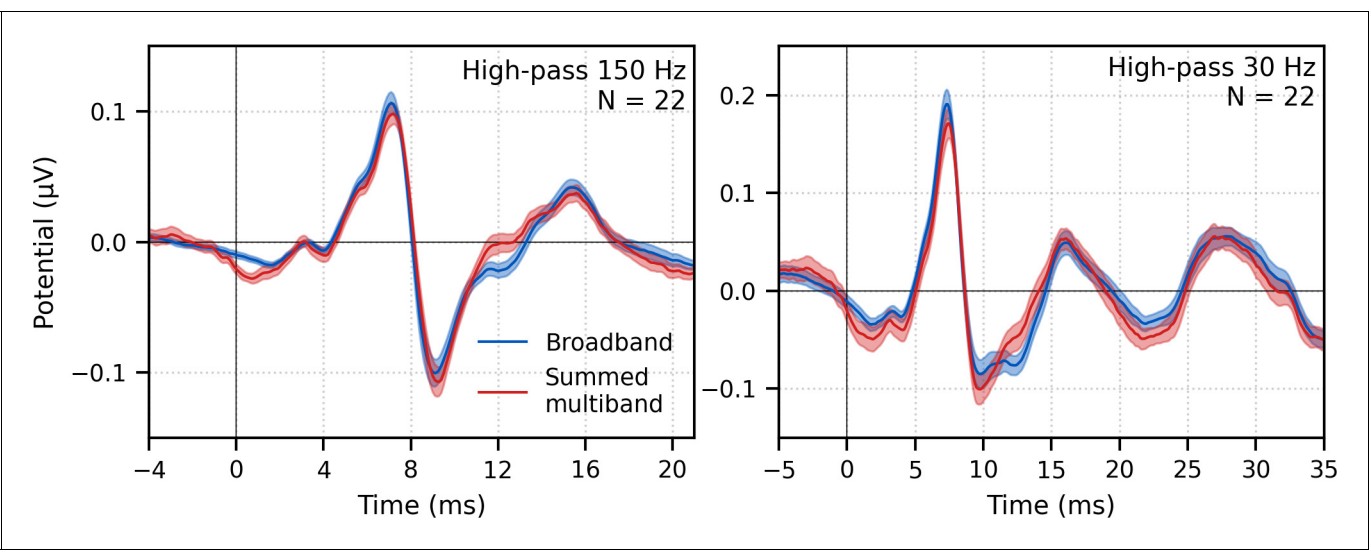

**Figure 7.** Comparison of responses to ~43 min of male-narrated peaky speech in the same subjects. Average waveforms across subjects (areas show ±1 SEM) are shown for broadband peaky speech (blue) and summed frequency-specific responses to multiband peaky speech with the common component added (red), high-pass filtered at 150 Hz (left) and 30 Hz (right). Regressors in the deconvolution were pulse trains.

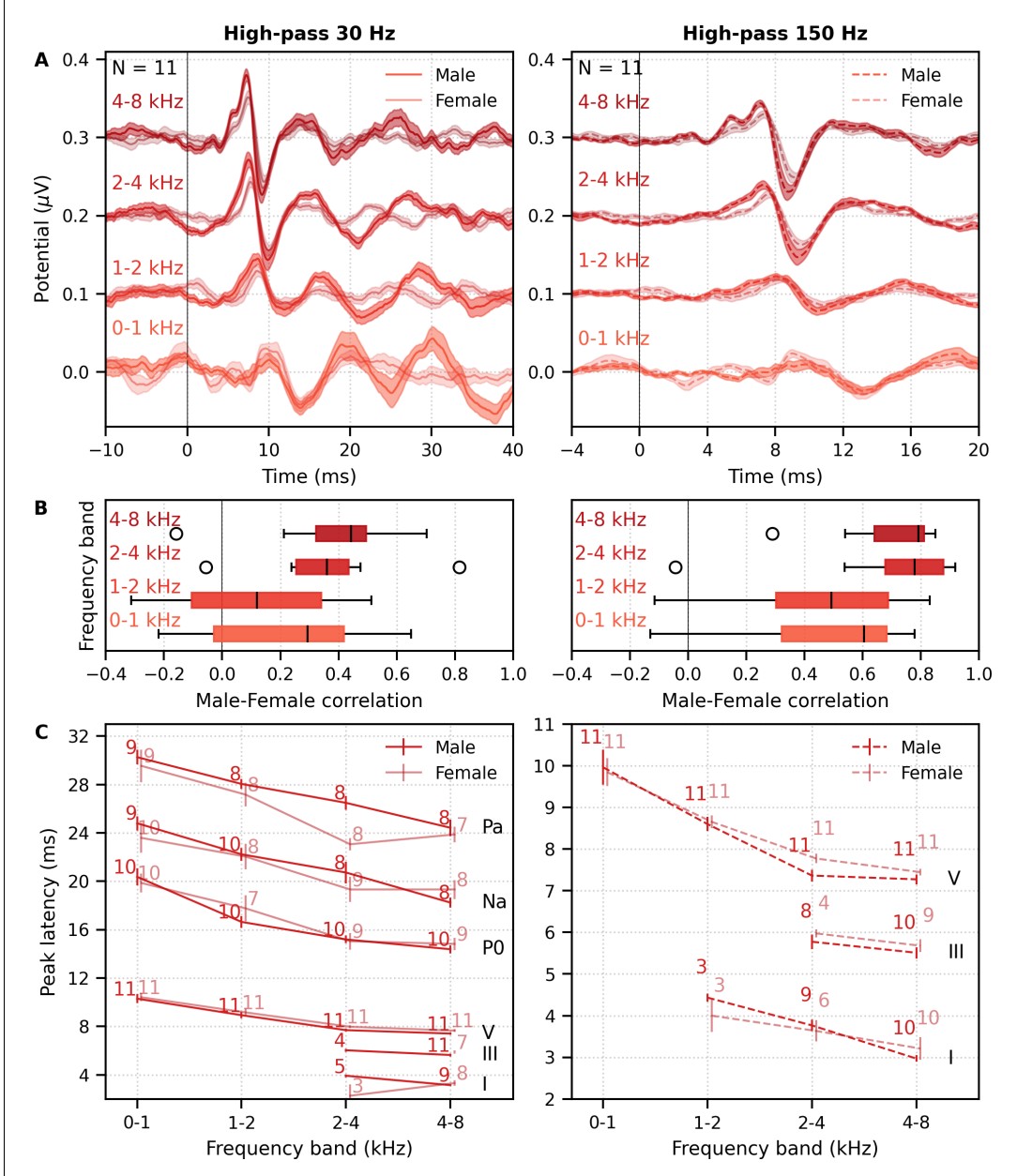

**Figure 8.** Comparison of responses to 32 min each of male- and female-narrated re-synthesized multiband peaky speech. (A) Average frequency-specific waveforms across subjects (areas show ±1 SEM; common component removed) are shown for each band in response to male- (dark red lines) and female-narrated (light red lines) speech. Responses were high-pass filtered at 30 Hz (left) and 150 Hz (right) to highlight the middle latency response (MLR) and auditory brainstem response (ABR), respectively. (B) Correlation coefficients between responses evoked by male- and female-narrated multiband peaky speech during ABR/MLR (left) and ABR (right) time lags for each frequency band. Black lines denote the median. (C) Mean ± SEM peak latencies for male- (dark) and female-narrated (light) speech for each wave decreased with increasing frequency band. Numbers of subjects with an identifiable wave are given for each wave, band, and narrator. Lines are given a slight horizontal offset to make the error bars easier to see. Details of the mixed effects models for (C) are provided in *Supplementary file 1B*.

for the 4–8 kHz band for MLR lags (*Figure 8B*, left), and from 0.49 (0.30–0.69) for the 1–2 kHz band to 0.79 (0.64–0.81) for the 4–8 kHz band for ABR lags (*Figure 8B*, right). These male−female correlation coefficients were significantly weaker than those of the same EEG split into even and odd trials for all but the 2–4 kHz frequency band when responses were high-pass filtered at 30 Hz and correlated across 0–40 ms lags (2–4 kHz: $W_{(10)} = 17.0$, p=0.175; other bands: $W_{(10)} \leq 5.0$, p≤0.010), but were similar to the even/odd trials for responses from all frequency bands high-pass filtered at 150

Hz ($W_{(10)} \geq 11.0$, p$\geq$0.054). These results indicate that the specific narrator can affect the robustness of frequency-specific responses, particularly for the MLR waves.

As expected from the grand average waveforms and male–female correlations, there were fewer subjects who had identifiable waves across frequency bands for the female- than male-narrated speech. These numbers are shown in *Figure 8C*, along with the mean ± SEM peak latencies for each wave, frequency band, and narrator. Again, there were few numbers of subjects with identifiable waves I and III for the lower frequency bands. Therefore, a similar power law model as used above was performed for waves V, $P_0$, $N_a$, and $P_a$ of responses in the four frequency bands that were high-pass filtered at 30 Hz. This model also included fixed effects of narrator and its associated interactions. Details of the statistical model are described in *Supplementary file 1B*. The post-cochlear delay estimate for the female narrator of 5.58 ms was slightly outside the normal range of 4.7–5.0 ms, which is likely a result of high-passing at 30 Hz here compared to 100 Hz in the previous studies. All other values were consistent with their norms and appear in *Table 1B*. As before, peak latencies were different for each wave and decreased with increasing frequency in a manner more pronounced for wave V than later waves (all p<0.001). There was a main effect of narrator on peak latencies but no interaction with wave (narrator p=0.001, wave–narrator interactions p>0.087). Therefore, as with broadband peaky speech, frequency-specific peaky responses were more robust with the male narrator and the frequency-specific responses peaked earlier for a narrator with a lower fundamental frequency.

## Frequency-specific responses can be measured simultaneously in each ear (dichotically)

We have so far demonstrated the effectiveness of peaky speech for diotic stimuli, but there is often a need to evaluate auditory function in each ear, and the most efficient tests assess both ears simultaneously. Applying this principle to generate multiband peaky speech, we investigated whether ear-specific responses could be evoked across the five standard audiological octave frequency bands (500–8000 Hz) using dichotic multiband speech. We created 10 independent pulse trains, two for each ear in each of the five frequency bands (see 'Multiband peaky speech' and 'Band filters' sections in 'Materials and methods').

We recorded responses to male- and female-narrated dichotic multiband peaky speech in 11 subjects. The frequency-specific (i.e., common component-subtracted) group average waveforms for each ear and frequency band are shown in *Figure 9A*. The 10 waveforms were small, especially for female-narrated speech, but a wave V was identifiable for both narrators. Also, waves I and III of the ABR were visible in the group average waveforms of the 4 kHz band ($\geq$45% and 18% of subjects for the male and female narrator, respectively) and 8 kHz band ($\geq$90% and 72% of subjects for the male and female narrator, respectively). MLR waves were not clearly identifiable for responses to female-narrated speech. Therefore, correlations between responses were performed for ABR lags between 0–15 ms. As shown in *Figure 9B*, the median (interquartile range) left–right ear correlation coefficients (averaged across narrators) ranged from 0.28 (0.02–0.52) for the 0.5 kHz band to 0.73 (0.62–0.84) for the 8 kHz band. Male–female correlation coefficients (averaged across ear) ranged from 0.44 (0.00–0.58) for the 0.5 kHz band to 0.76 (0.51–0.80) for the 8 kHz band. Although the female-narrated responses were smaller than the male-narrated responses, these male–female coefficients did not significantly differ from correlations of same EEG split into even–odd trials and averaged across ear ($W_{(10)} \geq 12.0$, p$\geq$0.067), likely reflecting the variability in such small responses.

*Figure 9C* shows the mean ± SEM peak latencies of wave V for each ear and frequency band for the male- and female-narrated dichotic multiband peaky speech. The change in wave V latency with frequency was modeled by a similar power law as used above. This model also included fixed effects of narrator, ear, and the interactions between narrator and frequency. Details of the statistical model are described in *Supplementary file 1C*. For wave V, the estimated mean parameters for both narrators were consistent with norms and appear in *Table 1C*. Latency decreased with increasing frequency (slope, p<0.001) in a similar way for both narrators (interaction with slope p=0.085). Peak latency differed by narrator (interaction with intercept p=0.004) but not between ears (p=0.265). Taken together, these results confirm that, while small in amplitude, frequency-specific responses can be elicited in both ears across five different frequency bands and show characteristic latency changes across the different frequency bands.

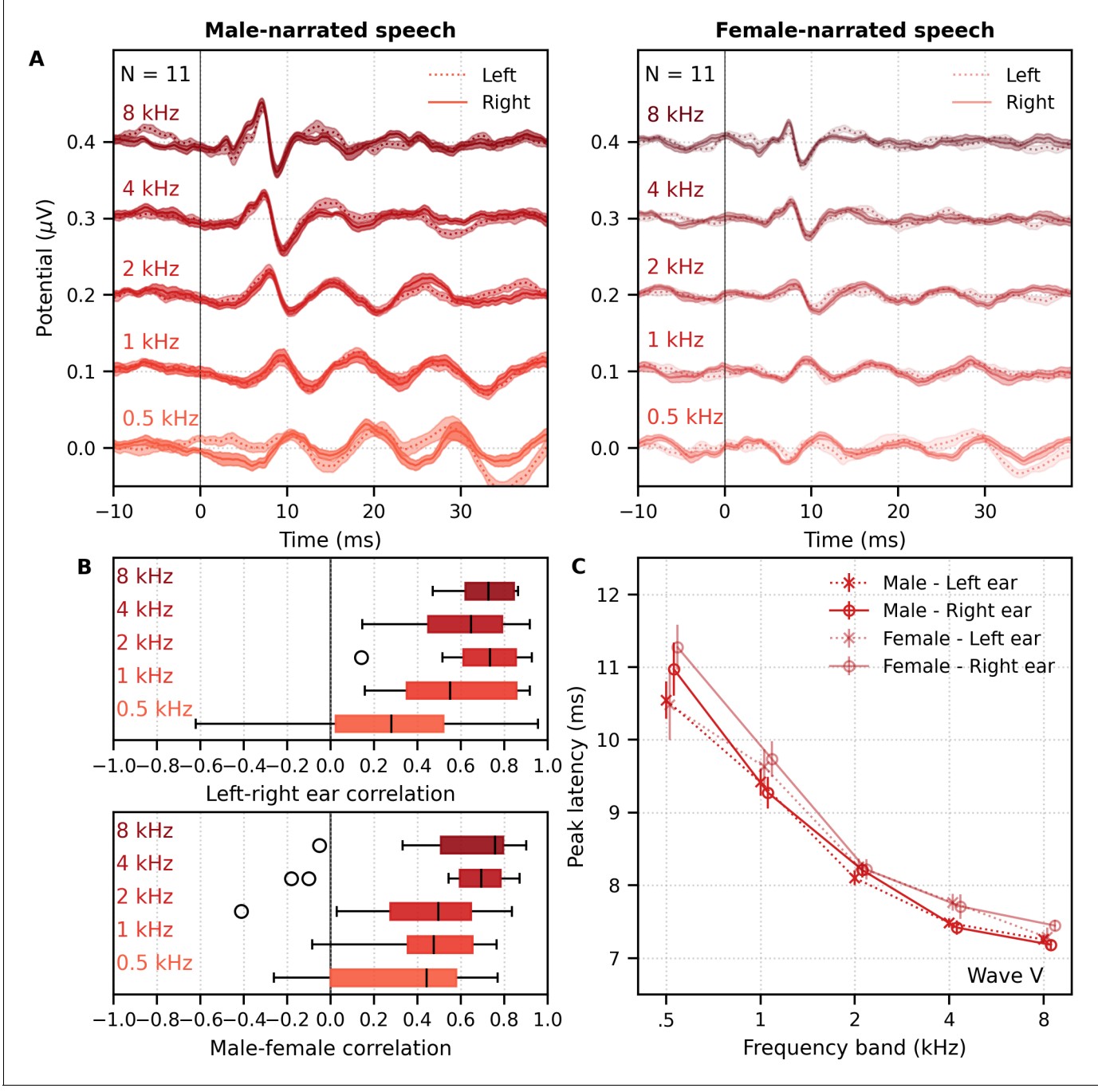

**Figure 9.** Comparison of responses to ~60 min each of male- and female-narrated dichotic multiband peaky speech with standard audiological frequency bands. (**A**) Average frequency-specific waveforms across subjects (areas show ±1 SEM; common component removed) are shown for each band for the left ear (dotted lines) and right ear (solid lines). Responses were high-pass filtered at 30 Hz. (**B**) Left–right ear correlation coefficients (top, averaged across gender) and male–female correlation coefficients (bottom, averaged across ear) during auditory brainstem response time lags (0–15 ms) for each frequency band. Black lines denote the median. (**C**) Mean ± SEM wave V latencies for male- (dark red) and female-narrated (light red) speech for the left (dotted line, cross symbol) and right ear (solid line, circle symbol) decreased with increasing frequency band. Lines are given a slight horizontal offset to make the error bars easier to see. Details of the mixed effects model for (**C**) are provided in *Supplementary file 1C*.

## Responses are obtained quickly for male-narrated broadband peaky speech but not multiband peaky speech

Having demonstrated that peaky broadband and multiband speech provides canonical waveforms with characteristic changes in latency with frequency, we next evaluated the acquisition time required for waveforms to reach a decent SNR (see 'Response SNR calculation' section in 'Materials and methods' for details).

*Figure 10* shows the cumulative proportion of subjects who had responses with ≥0 dB SNR to unaltered and broadband peaky speech as a function of recording time. Acquisition times for 22 subjects were similar for responses to both unaltered and broadband peaky male-narrated speech, with 0 dB SNR achieved by 7–8 min in 50% of subjects and about 20 min for all subjects. For responses high-pass filtered at 150 Hz to highlight the ABR (0–15 ms interval), the time reduced to 2 and 5 min for 50% and 100% of subjects respectively for broadband peaky speech but increased to 8 and >20 min for 50% and 100% of subjects respectively for unaltered speech. The increased time for the unaltered speech reflects the broad morphology of the response during ABR lags. These times for male-narrated broadband peaky speech were confirmed in our second cohort of 11 subjects. However, acquisition times were at least 2.1 times – but in some cases over 10 times – longer for female-narrated broadband peaky speech. In contrast to male-narrated speech, not all subjects achieved this threshold for female-narrated speech by the end of the 32 min recording. Taken together, these acquisition times confirm that responses with useful SNRs can be measured quickly for male-narrated broadband peaky speech but longer recording sessions are necessary for narrators with higher fundamental frequencies.

The longer recording times necessary for a female narrator became more pronounced for the multiband peaky speech. *Figure 11A* shows the cumulative density function for responses high-pass filtered at 150 Hz and the SNR estimated over the ABR interval. Many subjects (72%) had frequency-specific responses (common component subtracted) with ≥0 dB SNR for all four frequency bands by the end of the 32 min recording for the male-narrated speech, but this was achieved in only 45% of subjects for the female-narrated speech. Multiband peaky speech required significantly longer recording times than broadband peaky speech, with 50% of subjects achieving 0 dB SNR by 15 min

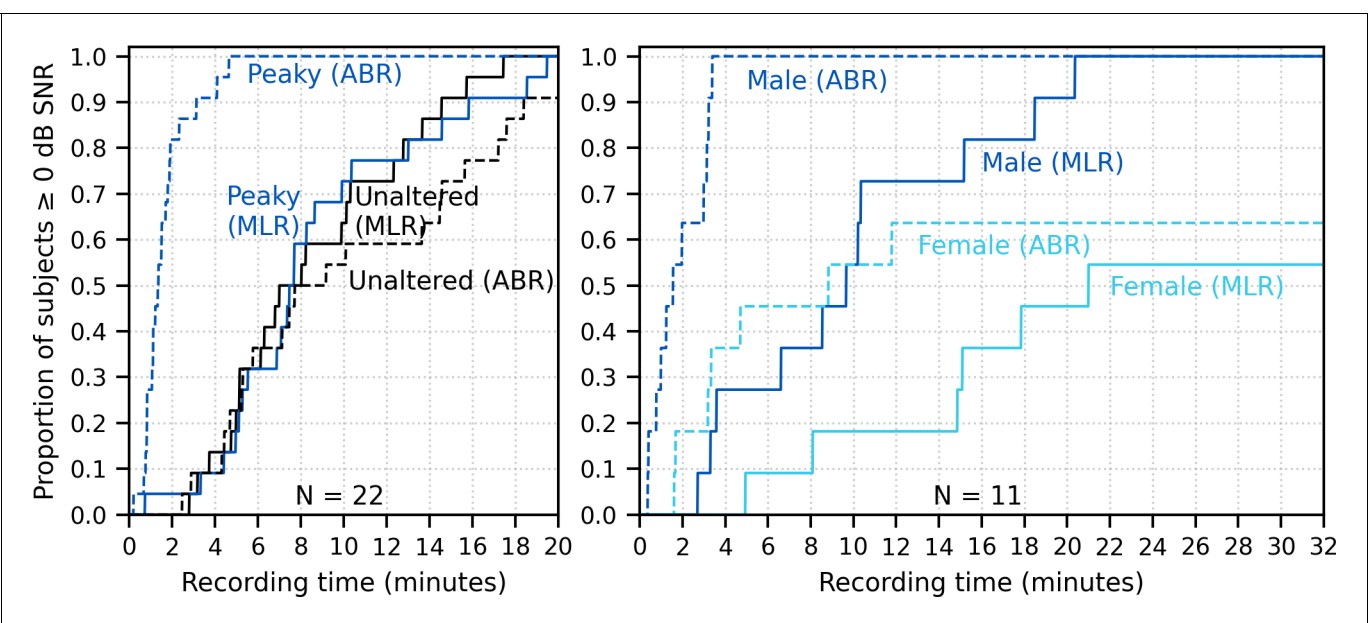

**Figure 10.** Cumulative proportion of subjects who have responses with ≥0 dB signal-to-noise ratio (SNR) as a function of recording time. Time required for unaltered (black) and broadband peaky speech (dark blue) of a male narrator is shown for 22 subjects in the left plot, and for male (dark blue) and female (light blue) broadband peaky speech is shown for 11 subjects in the right plot. Solid lines denote SNRs calculated using variance of the signal high-pass filtered at 30 Hz over the auditory brainstem response (ABR)/middle latency response (MLR) interval 0–30 ms, and dashed lines denote SNR variances calculated on signals high-pass filtered at 150 Hz over the ABR interval 0–15 ms. Noise variance was calculated in the pre-stimulus interval −480 to −20 ms.

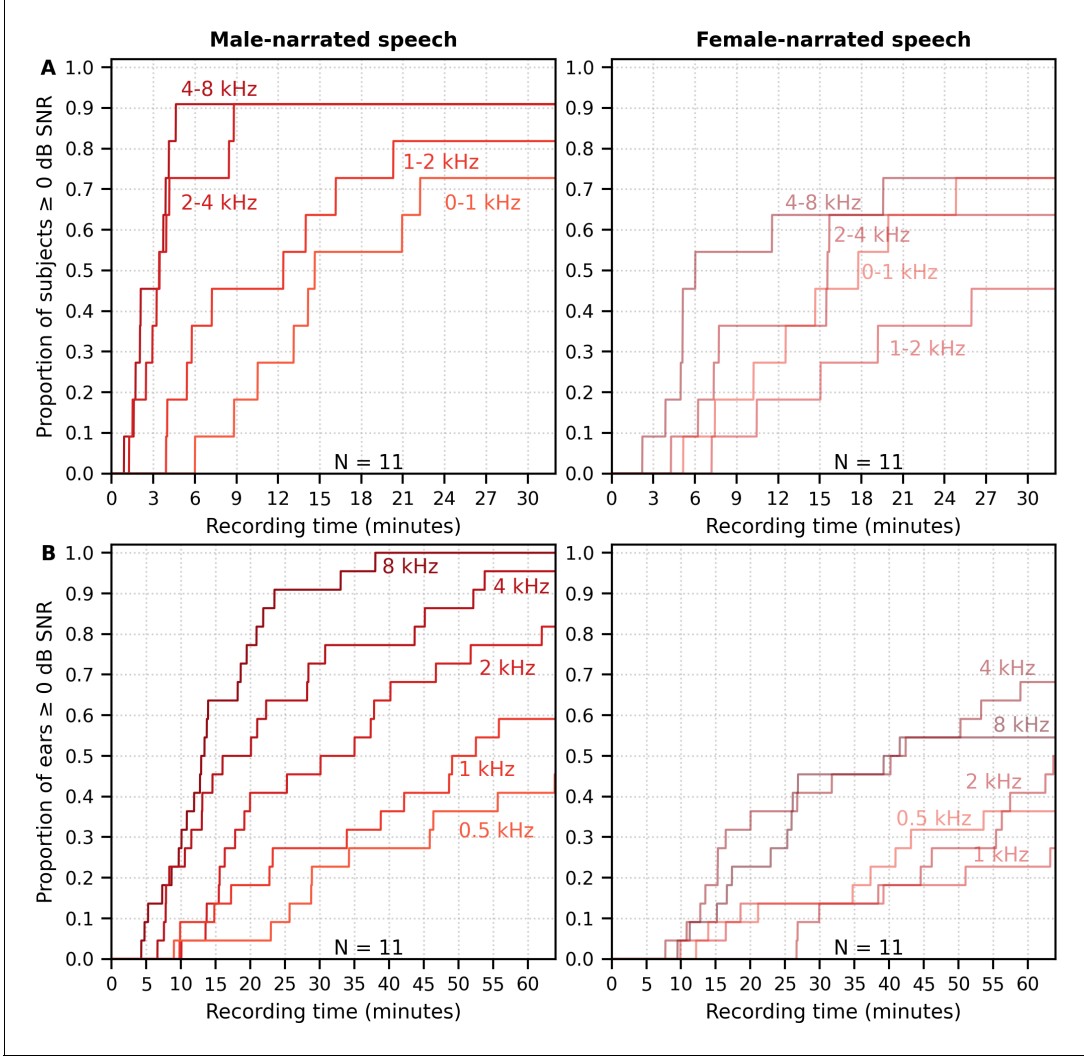

**Figure 11.** Cumulative proportion of subjects who have frequency-specific responses (common component subtracted) with ≥0 dB signal-to-noise ratio (SNR) as a function of recording time. Acquisition time was faster for male- (left) than female-narrated (right) multiband peaky speech with (**A**) four frequency bands presented diotically and (**B**) five frequency bands presented dichotically (total of 10 responses, five bands in each ear). SNR was calculated by comparing variance of signals high-pass filtered at 150 Hz across the auditory brainstem response interval of 0–15 ms to variance of noise in the pre-stimulus interval −480 to −20 ms.

compared to 2 min for the male-narrated responses across the ABR 0–15 ms interval and 17 min compared to 5 min for the MLR 0–30 ms interval. Even more time was required for dichotic multiband speech, which comprised a larger number of frequency bands (*Figure 11B*). All 10 audiological band responses achieved ≥ 0 dB SNR in 45% of ears (10/22 ears from 11 subjects) by 64 min for male-narrated speech and in 27% of ears (6/22 ears) for female-narrated speech. The smaller and broader responses in the low-frequency bands were slower to obtain and were the main constraint on testing time. These recording times suggest that deriving multiple frequency-specific responses will require at least more than 30 min per condition for <5 bands and more than an hour session for one condition of peaky multiband speech with 10 bands.

## Discussion

The major goal of this work was to develop a method to investigate early stages of naturalistic speech processing. We re-synthesized continuous speech taken from audiobooks so that the phases of all harmonics aligned at each glottal pulse during voiced segments, thereby making speech as

impulse-like (peaky) as possible to drive the auditory brainstem. Then we used the glottal pulse trains as the regressor in deconvolution to derive the responses. Indeed, comparing waveforms to broadband peaky and unaltered speech validated the superior ability of peaky speech to evoke additional waves of the canonical ABR and MLR, reflecting neural activity from multiple subcortical structures. Robust ABR and MLR responses were recorded in less than 5 and 20 min, respectively, for all subjects, with half of the subjects exhibiting a strong ABR within 2 min and MLR within 8 min. Longer recording times were required for the smaller responses generated by a narrator with a higher fundamental frequency. We also demonstrated the flexibility of this stimulus paradigm by simultaneously recording up to 10 frequency-specific responses to multiband peaky speech that was presented either diotically or dichotically, although these responses required much longer recording times. Taken together, our results show that peaky speech effectively yields responses from distinct subcortical structures and different frequency bands, paving the way for new investigations of speech processing and new tools for clinical application.

## Peaky speech responses reflect activity from distinct subcortical components

### Canonical responses can be derived from speech with impulse-like characteristics

For the purpose of investigating responses from different subcortical structures, we accomplished our goal of creating a stimulus paradigm that overcame some of the limitations of current methods using natural speech. Methods that do not use re-synthesized impulse-like speech generate responses characterized by a broad peak between 6–9 ms (*Forte et al., 2017*; *Maddox and Lee, 2018*), with contributions predominantly from the inferior colliculus (*Saiz-Alía and Reichenbach, 2020*). In contrast, for the majority of our subjects, peaky speech evoked responses with canonical morphology comprising waves I, III, V, $P_0$, $N_a$, and $P_a$ (*Figure 1*), reflecting neural activity from distinct stages of the auditory system from the auditory nerve to thalamus and primary auditory cortex (e.g., *Picton et al., 1974*). Although clicks also evoke responses with multiple component waves, current studies of synaptopathy show quite varied results with clicks and poor correlation with speech in noise (*Bramhall et al., 2019*; *Prendergast et al., 2017*). Obtaining click-like responses to stimuli with all of speech's spectrotemporal richness may provide a better connection to the specific neural underpinnings of speech encoding, similar to how the complex cross-correlation to a fundamental waveform can change based on the context of attention (*Forte et al., 2017*; *Saiz-Alía et al., 2019*).

The ABR wave V and similar MLR waves evoked here were also evoked by a method using embedded chirps intermixed within alternating octave bands of speech (*Backer et al., 2019*). Chirps are transients that compensate for the cochlear traveling delay wave by introducing different phases across frequency, leading to a more synchronized response across the cochlea and a larger brainstem response than for clicks (*Dau et al., 2000*; *Elberling and Don, 2008*; *Shore and Nuttall, 1985*). The responses to embedded chirps elicited waves with larger mean amplitude than those to our broadband peaky speech (~0.4 versus ~0.2 μV, respectively), although a similar proportion of subjects had identifiable waves, the SNR was good, and several other factors may contribute to amplitude differences. For example, higher click rates (e.g., *Burkard et al., 1990*; *Burkard and Hecox, 1983*; *Chiappa et al., 1979*; *Don et al., 1977*; *Jiang et al., 2009*) and higher fundamental frequencies (*Maddox and Lee, 2018*; *Saiz-Alía et al., 2019*; *Saiz-Alía and Reichenbach, 2020*) reduce the brainstem response amplitude, and dynamic changes in rate may create interactions across neural populations that lead to smaller amplitudes. Our stimuli kept the dynamic changes in pitch across all frequencies (instead of alternate octave bands of chirps and speech) and created impulses at every glottal pulse, with an average pitch of ~115 Hz and ~198 Hz for the male and female narrators, respectively. These presentation rates were much higher and more variable than the flat 42 Hz rate at which the embedded chirps were presented (pitch flattened to 82 Hz and chirps presented every other glottal pulse). We could evaluate whether chirps would improve response amplitude to our dynamic peaky speech by simply all-pass filtering the re-synthesized voiced segments by convolving with a chirp prior to mixing the re-synthesized parts with the unvoiced segments. While maintaining the amplitude spectrum of speech, the harmonics would then have the different phases associated with chirps at each glottal pulse instead of all phases set to 0.

Regardless, our peaky speech generated robust canonical responses with good SNR while maintaining a natural-sounding, if very slightly 'buzzy', quality to the speech. Overall, continuous speech re-synthesized to contain impulse-like characteristics is an effective way to elicit responses that distinguish contributions from different subcortical structures.

## Latencies of component waves are consistent with distinct subcortical structures

The latencies of the component waves of the responses to peaky speech are consistent with activity arising from known subcortical structures. The inter-wave latencies between I–III, III–V, and I–V fall within the expected range for brainstem responses elicited by transients at 50–60 dB SL and 50–100 Hz rates (*Burkard and Hecox, 1983*; *Chiappa et al., 1979*; *Don et al., 1977*), suggesting that the transmission times between auditory nerve, cochlear nucleus, and rostral brainstem remain similar for speech stimuli. However, these speech-evoked waves peak at later absolute latencies than responses to transient stimuli at 60 dB SL and 90–100 Hz, but at latencies more similar to those presented at 50 dB SL or 50 dB nHL in the presence of some masking noise (*Backer et al., 2019*; *Burkard and Hecox, 1983*; *Chiappa et al., 1979*; *Don et al., 1977*; *Maddox and Lee, 2018*). There are reasons why the speech-evoked latencies may be later. First, our level of 65 dB sound pressure level (SPL) may be more similar to click levels of 50 dB SL. Second, although spectra of both speech and transients are broad, clicks, chirps, and even our previous speech stimuli (which was high-pass filtered at 1 kHz; *Maddox and Lee, 2018*) have relatively greater high-frequency energy than the unaltered and peaky broadband speech used in the present work. Neurons with higher characteristic frequencies respond earlier due to their basal cochlear location and contribute relatively more to brainstem responses (e.g., *Abdala and Folsom, 1995*), leading to quicker latencies for stimuli that have greater high-frequency energy. Also consistent with having greater lower frequency energy, our unaltered and peaky speech responses were later than the response from the same speech segments that were high-pass filtered at 1 kHz (*Maddox and Lee, 2018*). In fact, the ABR to broadband peaky speech bore a close resemblance to the summation of each frequency-specific response and the common component to peaky multiband speech (*Figure 7*), with peak wave latencies representing the relative contribution of each frequency band. Third, higher stimulation rates prolong latencies due to neural adaptation, and the 115–198 Hz average fundamental frequencies of our speech were much higher than the 41 Hz embedded chirps and 50–100 Hz click rates (e.g., *Burkard et al., 1990*; *Burkard and Hecox, 1983*; *Chiappa et al., 1979*; *Don et al., 1977*; *Jiang et al., 2009*). The effect of stimulation rate was also demonstrated by the later ABR wave I, III, and V peak latencies for the female narrator with the higher average fundamental frequency of 198 Hz (*Figure 6A, C*). Therefore, the differing characteristics of typical periodic transients (such as clicks and chirps) and continuous speech may give rise to differences in brainstem responses, even though they share canonical waveforms arising from similar contributing subcortical structures.

The latency of the peaky speech-evoked response also differed from the non-standard, broad responses to unaltered speech. However, latencies from these waveforms are difficult to compare due to the differing morphology and the different analyses that were used to derive the responses. Evidence for the effect of analysis comes from the fact that the same EEG collected in response to peaky speech could be regressed with pulse trains to give canonical ABRs (*Figure 1*, *Figure 2*), or regressed with the half-wave rectified peaky speech to give the different, broad waveform (*Figure 4*). Furthermore, non-peaky continuous speech stimuli with similar ranges of fundamental frequencies (between 100–300 Hz) evoke non-standard, broad brainstem responses that also differ in morphology and latency depending on whether the EEG is analyzed by deconvolution with the half-wave rectified speech (*Figure 2*, *Maddox and Lee, 2018*) or complex cross-correlation with the fundamental frequency waveform (*Forte et al., 2017*). Therefore, again, even though the inferior colliculus and lateral lemniscus may contribute to generating these different responses (*Møller and Jannetta, 1983*; *Saiz-Alía and Reichenbach, 2020*; *Starr and Hamilton, 1976*), the morphology and latency may differ (sometimes substantially) depending on the analysis technique used.

## Responses reflect activity from different frequency regions

In addition to evoking canonical brainstem responses, peaky speech can be exploited for other traditional uses of ABR, such as investigating subcortical responses across different frequencies.

Frequency-specific responses were measurable to two different types of multiband peaky speech: four frequency bands presented diotically (*Figure 6*, *Figure 8*) and five frequency bands presented dichotically (*Figure 9*). Peak wave latencies of these responses decreased with increasing band frequency in a similar way to responses evoked by tone pips and derived bands from clicks in noise (*Gorga et al., 1988*; *Neely et al., 1988*; *Rasetshwane et al., 2013*; *Strelcyk et al., 2009*), thereby representing activity evoked from different areas across the cochlea. In fact, our estimates of the power law parameters of *a* (the central conduction time), *d* (the frequency dependence), and *b* (the latency corresponding to 1 kHz and 65 dB SPL) for wave V fell within the corresponding ranges that were previously reported for tone pips and derived bands at 65 dB ppeSPL (*Neely et al., 1988*; *Rasetshwane et al., 2013*; *Strelcyk et al., 2009*). Interestingly, the frequency-specific responses across frequency band were similar in amplitude or possibly slightly smaller for the lower frequency bands (*Figure 6*, *Figure 8*, *Figure 9*), even though the relative energy of each band decreased with increasing frequency, resulting in an ~30 dB difference between the lowest and highest frequency bands (Figure 14). A greater response elicited by higher frequency bands is consistent with the relatively greater contribution of neurons with higher characteristic frequencies to ABRs (*Abdala and Folsom, 1995*), as well as the need for higher levels to elicit low-frequency responses to tone pips that are close to threshold (*Gorga et al., 2006*; *Gorga et al., 1993*; *Hyde, 2008*; *Stapells and Oates, 1997*). Also, canonical waveforms were derived in the higher frequency bands of diotically presented speech, with waves I and III identifiable in most subjects. Multiband peaky speech will not replace the current frequency-specific ABR, but there are situations where it may be advantageous to use speech over tone pips. Measuring waves I, III, and V of high-frequency responses in the context of all the dynamics of speech may have applications to studying effects of cochlear synaptopathy on speech comprehension (*Bharadwaj et al., 2014*; *Liberman et al., 2016*). Another exciting potential application is the evaluation of supra-threshold hearing across frequency in toddlers and individuals who do not provide reliable behavioral responses as they may be more responsive to sitting for longer periods of time while listening to a narrated story than to a series of tone pips. Giving a squirmy toddler an iPad and presenting a story for 30 min could allow responses to multiband peaky speech that confirm audibility at normal speech levels. Such a screening or metric of audibility is a useful piece of knowledge in pediatric clinical management and is also being investigated for the frequency following response (*Easwar et al., 2020*; *Easwar et al., 2015*), which provides different information than the canonical responses. An extension of this assessment would be to evaluate neural speech processing in the context of hearing loss, as well as rehabilitation strategies such as hearing aids and cochlear implants. Auditory prostheses have algorithms specifically tuned for the spectrotemporal dynamics of speech that behave very differently in response to standard diagnostic stimuli such as trains of clicks or tone pips. Peaky speech responses could allow us to assess how the auditory system is encoding the amplified speech and validate audibility of the hearing aid fittings before the infant or toddler is old enough to provide reliable speech perception testing. Therefore, the ability of peaky speech to yield both canonical waveforms and frequency-specific responses makes this paradigm a flexible method that assesses speech encoding in new ways.

## Practical considerations for using peaky speech and deconvolution

### Filtering

Having established that peaky speech is a flexible stimulus for investigating different aspects of speech processing, there are several practical considerations for using the peaky speech paradigm. First, filtering should be performed carefully. As recommended in *Maddox and Lee, 2018*, causal filters – which have impulse responses with non-zero values at positive lags only – should be used to ensure cortical activity at later peak latencies does not spuriously influence earlier peaks corresponding to subcortical origins. Applying less aggressive, low-order filters (i.e., broadband with shallow roll-offs) will help reduce the effects of causal filtering on delaying response latency. The choice of high-pass cutoff will also affect the response amplitude and morphology. After evaluating several orders and cutoffs to the high-pass filters, we determined that early waves of the peaky broadband ABRs were best visualized with a 150 Hz cutoff, whereas a lower cutoff frequency of 30 Hz was necessary to view the ABR and MLR of the broadband responses. When evaluating specific contributions to the earliest waves, we recommend at least a first-order 150 Hz high-pass filter or a more aggressive second-order 200 Hz high-pass filter to deal with artifacts arising from nonlinearities that

are not taken into account by the pulse train regressor or any potential influences by responses from later sources (*Figure 3*). For multiband responses, the 150 Hz high-pass filter significantly reduced the response but also decreased the low-frequency noise in the pre-stimulus interval. For the 4-band multiband peaky speech, the 150 Hz and 30 Hz filters provided similar acquisition times for 0 dB SNR, but better SNRs were obtained quicker with 150 Hz filtering for the 10-band multiband peaky speech.

## Choice of narrator (stimulation rate)

Second, the choice of narrator impacts the responses to both broadband and multiband peaky speech. Although overall morphology was similar, the male-narrated responses were larger, contained more clearly identifiable component waves in a greater proportion of subjects, and achieved a 0 dB SNR at least 2.1 to over 10 times faster than those evoked by a female narrator. These differences likely stemmed from the ~77 Hz difference in average pitch as higher stimulation rates evoke smaller responses due to adaptation and refractoriness (e.g., *Burkard et al., 1990*; *Burkard and Hecox, 1983*; *Chiappa et al., 1979*; *Don et al., 1977*; *Jiang et al., 2009*). Indeed, a 50 Hz change in fundamental frequency yields a 24% reduction in the modeled ABR that was derived as the complex cross-correlation with the fundamental frequency (*Saiz-Alía and Reichenbach, 2020*). The narrator differences exhibited in the present study may be larger than those in other studies with continuous speech (*Forte et al., 2017*; *Maddox and Lee, 2018*; *Saiz-Alía et al., 2019*) as a result of the different regressors. These response differences do not preclude using narrators with higher fundamental frequencies in future studies, but the time required for usable responses from each narrator must be considered when planning experiments, and caution taken when interpreting comparisons between conditions with differing narrators. The strongest results will come from comparing responses to the same narrator (or even the same speech recordings) under different experimental conditions.

## SNR and recording time for multiple responses

Third, the necessary recording time depends on the chosen SNR threshold, experimental demands, and stimulus. We chose a threshold SNR of 0 dB based on when waveforms became clearly identifiable, but of course a different threshold would change our recording time estimates (though, notably, not the ratios between them). With this SNR threshold, acquisition times were quick enough for broadband peaky responses to allow multiple conditions in a reasonable recording session. With male-narrated broadband peaky speech, all subjects achieved 0 dB SNR ABRs in <5 min and MLRs in <20 min, thereby affording between 3 and 12 conditions in an hour recording session. These recording times are comparable, if not faster, than the 8 min for the broad response to unaltered speech, 6–12 min for the chirp-embedded speech (*Backer et al., 2019*), ~10 min for the broad complex-cross correlation response to the fundamental waveform (*Forte et al., 2017*), and 33 min for the broad response to high-passed continuous speech (*Maddox and Lee, 2018*). However, using a narrator with a higher fundamental frequency could increase testing time by two- to over tenfold. In this experiment, at most two conditions per hour could be tested with the female-narrated broadband peaky speech. Unlike broadband peaky speech, the testing times required for all frequency-specific responses to reach 0 dB SNR were significantly longer, making only one condition feasible within a recording session. At least 30 min was necessary for the multiband peaky speech, but based on extrapolated testing times, about 1 hr is required for 90% of subjects and 2 hr for 75% of subjects to achieve this threshold for all 4 bands of diotic speech and all 10 bands of dichotic speech, respectively. It is also possible that there was some contralateral suppression in our dichotic recordings; however, it is unlikely that separate monaural presentation would enlarge responses enough to be worth the doubled recording time. The longer testing times here are important to consider when planning studies using multiband peaky speech with several frequency bands.

## Number of frequency bands

Fourth, as mentioned above, the number of frequency bands incorporated into multiband peaky speech decreases SNR and increases testing time. Although it is possible to simultaneously record up to 10 frequency-specific responses, the significant time required to obtain decent SNRs reduces the feasibility of testing multiple conditions or having recording sessions lasting less than 1–2 hr.

However, pursuing shorter testing times with multiband peaky speech is possible. Depending on the experimental question, different multiband options could be considered. For male-narrated speech, the 2–4 and 4–8 kHz responses had good SNRs and exhibited waves I, III, and V within 9 min for 90% of subjects. Therefore, if researchers were more interested in comparing responses in these higher frequency bands, they could stop recording once these bands reach threshold but before the lower frequency bands reach criterion (i.e., within 9 min). Alternatively, the lower frequencies could be combined into a single broader band in order to reduce the total number of bands or the intensity could be increased to evoke responses with larger amplitudes. Therefore, different band and parameter considerations could reduce testing time and improve the feasibility, and thus utility, of multiband peaky speech.

## Flexible analysis windows for deriving responses from auditory nerve to cortex

Fifth, and finally, a major advantage of deconvolution analysis is that the analysis window for the response can be extended arbitrarily in either direction to include a broader range of latencies (*Maddox and Lee, 2018*). Extending the pre-stimulus window leftward provides a better estimate of the SNR, and extending the window rightward allows parts of the response that come after the ABR and MLR to be analyzed as well, which are driven by the cortex. These later responses can be evaluated in response to broadband peaky speech, but as shown in *Figures 8* and *9*, only ABR and early MLR waves are present in the frequency-specific responses. The same broadband peaky speech data from *Figure 5* are high-pass filtered at 1 Hz and displayed with an extended time window in *Figure 12*, which shows component waves of the ABR, MLR, and late latency responses (LLR). Thus, this method allows us to simultaneously investigate speech processing ranging from the earliest level of the auditory nerve all the way through the cortex without requiring extra recording time. Usually the LLR is larger than the ABR/MLR, but our subjects were encouraged to relax and rest, yielding a passive LLR response. Awake and attentive subjects may improve the LLR; however, other studies that present continuous speech to attentive subjects also report smaller and different LLR (*Backer et al., 2019*; *Maddox and Lee, 2018*), possibly from cortical adaptation to a continuous stimulus. Here, we used a simple two-channel montage that is optimized for recording ABRs, but a full multichannel montage could also be used to more fully explore the interactions between subcortical and cortical processing of naturalistic speech. The potential for new knowledge about how the brain processes naturalistic and engaging stimuli cannot be undersold.

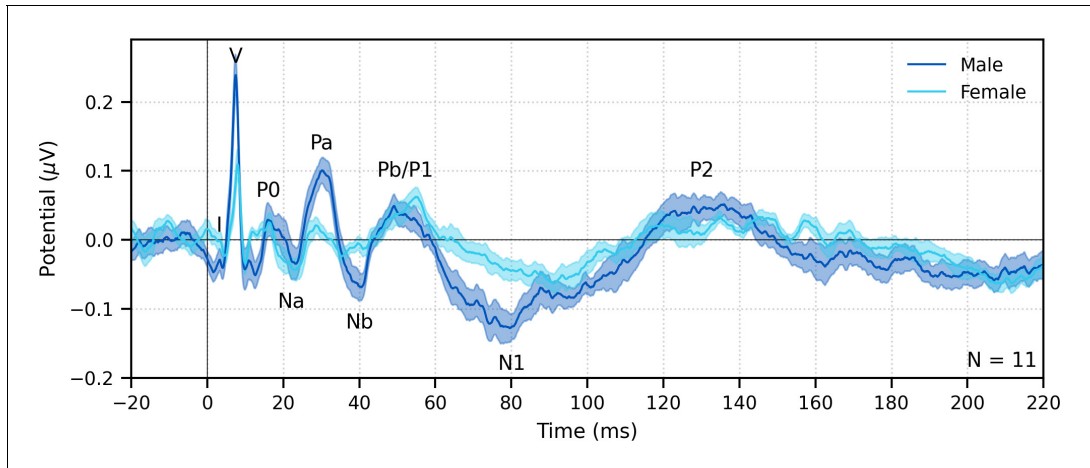

**Figure 12.** The range of lags can be extended to allow early, middle, and late latency responses to be analyzed from the same recording to broadband peaky speech. Average waveforms across subjects (areas show ±1 SEM) are shown for responses measured to 32 min of broadband peaky speech narrated by a male (dark blue) and female (light blue). Responses were high-pass filtered at 1 Hz using a first-order Butterworth filter, but different filter parameters can be used to focus on each stage of processing. Canonical waves of the auditory brainstem response, middle latency response, and late latency response are labeled for the male-narrated speech. Due to adaptation, amplitudes of the late potentials are smaller than typically seen with other stimuli that are shorter in duration with longer inter-stimulus intervals than our continuous speech. Waves I and III become more clearly visible by applying a 150 Hz high-pass cutoff.

### Peaky speech is a tool that opens up new lines of query

The peaky speech paradigm is a viable method for recording broadband and frequency-specific responses from distinct subcortical structures using an engaging, continuous speech stimulus. The customizability and flexibility of peaky speech facilitates new lines of query, both in neuroscientific and clinical domains. Speech often occurs within a mixture of sounds, such as other speech sources, background noise, or music. Furthermore, visual cues from a talker's face are often available to aid speech understanding, particularly in environments with low SNR (e.g., *Bernstein and Grant, 2009*; *Grant et al., 2007*). Peaky speech facilitates investigation into the complex subcortical encoding and processing that underpins successful listening in these scenarios using naturalistic, engaging tasks. Indeed, previous methods have been quite successful in elucidating cortical processing of speech under these conditions (*O'Sullivan et al., 2019*; *Teoh and Lalor, 2019*). Whereas these cortical studies could use regressors that are not acoustically based – such as semantics or surprisal – the fast responses of subcortical structures necessitate a regressor that can allow timing of components to separate subcortical from cortical origins. The similarity of the peaky speech response to the click response is an advantage because it is the only way to understand what is occurring in separate subcortical regions during the dynamic spectrotemporal context of speech. As with other work in this area, how informative the response is about speech processing will depend on how it is deployed experimentally. Experiments that measure the response to the same stimuli in different cognitive states (unattended/attended, understood/not understood) will illuminate the relationship between those states and subcortical encoding. Such an approach was recently used to show how the brainstem complex cross-correlation to a fundamental waveform can change based on top-down attention (*Forte et al., 2017*). Finally, the ability to customize peaky speech for measuring frequency-specific responses provides potential applications to clinical research in the context of facilitating assessment of supra-threshold hearing function and changes to how speech may be encoded following intervention strategies and technologies while using a speech stimulus that algorithms in hearing aids and cochlear implants are designed to process.

## Materials and methods

### Participants

Data were collected over three experiments that were conducted under a protocol approved by the University of Rochester Research Subjects Review Board (#1227). All subjects gave informed consent before the experiment began and were compensated for their time. In each of experiments 1 and 2, there were equipment problems during testing for one subject, rendering data unusable in the analyses. Therefore, there were a total of 22, 11, and 11 subjects included in experiments 1, 2, and 3 respectively. Four subjects completed both experiments 1 and 2, and two subjects completed both experiments 2 and 3. The 38 unique subjects (25 females, 66%) were aged 18–32 years with a mean ± SD age of 23.0 ± 3.6 years. Audiometric screening confirmed subjects had normal hearing in both ears, defined as thresholds ≤20 dB HL from 250 to 8000 Hz. All subjects identified English as their primary language.

### Stimulus presentation and EEG measurement

In each experiment, subjects listened to 128 min of continuous speech stimuli while reclined in a darkened sound booth. They were not required to attend to the speech and were encouraged to relax and sleep. Speech was presented at an average level of 65 dB SPL over ER-2 insert earphones (Etymotic Research, Elk Grove, IL) plugged into an RME Babyface Pro digital sound card (RME, Haimhausen, Germany) via an HB7 headphone amplifier (Tucker Davis Technologies, Alachua, FL). Stimulus presentation was controlled by a custom python (Python Programming Language, RRID: SCR_008394) script using publicly available software (Expyfun, RRID:SCR_019285; available at https://github.com/LABSN/expyfun; *Larson et al., 2014*). We interleaved conditions in order to prevent slow impedance drifts or transient periods of higher EEG noise from unevenly affecting one condition over the others. Physical measures to reduce stimulus artifact included (1) hanging earphones from the ceiling so that they were as far away from the EEG cap as possible and (2) sending an inverted signal to a dummy earphone (blocked tube) attached in the same physical orientation to the stimulus presentation earphones in order to cancel electromagnetic fields away from

transducers. The sound card also produced a digital signal at the start of each epoch, which was converted to trigger pulses through a custom trigger box (modified from a design by the National Acoustic Laboratories, Sydney, NSW, Australia) and sent to the EEG system so that audio and EEG data could be synchronized with sub-millisecond precision.

EEG was recorded using BrainVision's PyCorder software (RRID:SCR_019286). Ag/AgCl electrodes were placed at the high forehead (FCz, active non-inverting), left and right earlobes (A1, A2, inverting references), and the frontal pole (Fpz, ground). These were plugged into an EP-Preamp system specifically for recording ABRs, connected to an ActiCHamp recording system, both manufactured by BrainVision. Data were sampled at 10,000 Hz and high-pass filtered at 0.1 Hz. Offline, raw data were high-pass filtered at 1 Hz using a first-order causal Butterworth filter to remove slow drift in the signal, and then notch filtered with 5 Hz wide second-order infinite impulse response notch filters to remove 60 Hz and its first three odd harmonics (180, 300, 420 Hz). To optimize parameters for viewing the ABR and MLR components of peaky speech responses, we evaluated several orders and high-pass cutoffs to the filters. Early waves of the broadband peaky ABRs were best visualized with a 150 Hz cutoff, whereas a lower cutoff frequency of 30 Hz was necessary to view the ABR and MLR of the broadband responses. Conservative filtering with a first-order filter was sufficient with these cutoff frequencies.

## Speech stimuli and conditions

Speech stimuli were taken from two audiobooks. The first was *The Alchemyst* (*Scott, 2007*), read by a male narrator and used in all three experiments. The second was *A Wrinkle in Time* (*L'Engle, 2012*), read by a female narrator and used in experiments 2 and 3. The average fundamental frequency was 115 Hz for the male narrator and 198 Hz for the female narrator. These stimuli were used in *Maddox and Lee, 2018*, but in that study a gentle high-pass filter was applied, which was not done for this study. Briefly, the audiobooks were resampled to 44,100 Hz and then silent pauses were truncated to 0.5 s. Speech was segmented into 64 s epochs with 1 s raised cosine fade-in and fade-out. Because conditions were interleaved, the last 4 s of a segment were repeated in the next segment so that subjects could pick up where they left off if they were listening.

In experiment 1, subjects listened to three conditions of male speech (42.7 min each): unaltered speech, re-synthesized broadband peaky speech, and re-synthesized multiband peaky speech (see below for a description of re-synthesized speech). In experiment 2, subjects listened to four conditions of re-synthesized peaky speech (32 min each): male and female narrators of both broadband and multiband peaky speech. For these first two experiments, speech was presented diotically (same speech to both ears). In experiment 3, subjects listened to both male and female dichotic (slightly different stereo speech of the same narrator in each ear) multiband peaky speech designed for audiological applications (64 min of each narrator). The same 64 s of speech was presented simultaneously to each ear, but the stimuli were dichotic due to how the re-synthesized multiband speech was created (see below).

## Stimulus design

The brainstem responds best to impulse-like stimuli, so we re-synthesized the speech segments from the audiobooks (termed 'unaltered') to create three types of 'peaky' speech, with the objectives of (1) evoking additional waves of the ABR reflecting other neural generators and (2) measuring responses to different frequency regions of the speech. The process is described in detail below but is best read in tandem with the code that is publicly available (https://github.com/maddoxlab). *Figure 13* compares the unaltered speech and re-synthesized broadband and multiband peaky speech. Comparing the pressure waveforms shows that the peaky speech is as click-like as possible, but comparing the spectrograms (how sound varies in amplitude at every frequency and time point) shows that the overall spectrotemporal content that defines speech is basically unchanged by the re-synthesis. *Audio files 1– 6* provide examples of each stimulus type for both narrators, which demonstrate the barely perceptible difference between unaltered and peaky speech.

### Broadband peaky speech

Voiced speech comprises rapid openings and closings of the vocal folds, which are then filtered by the mouth and vocal tract to create different vowel and consonant sounds. The first processing step

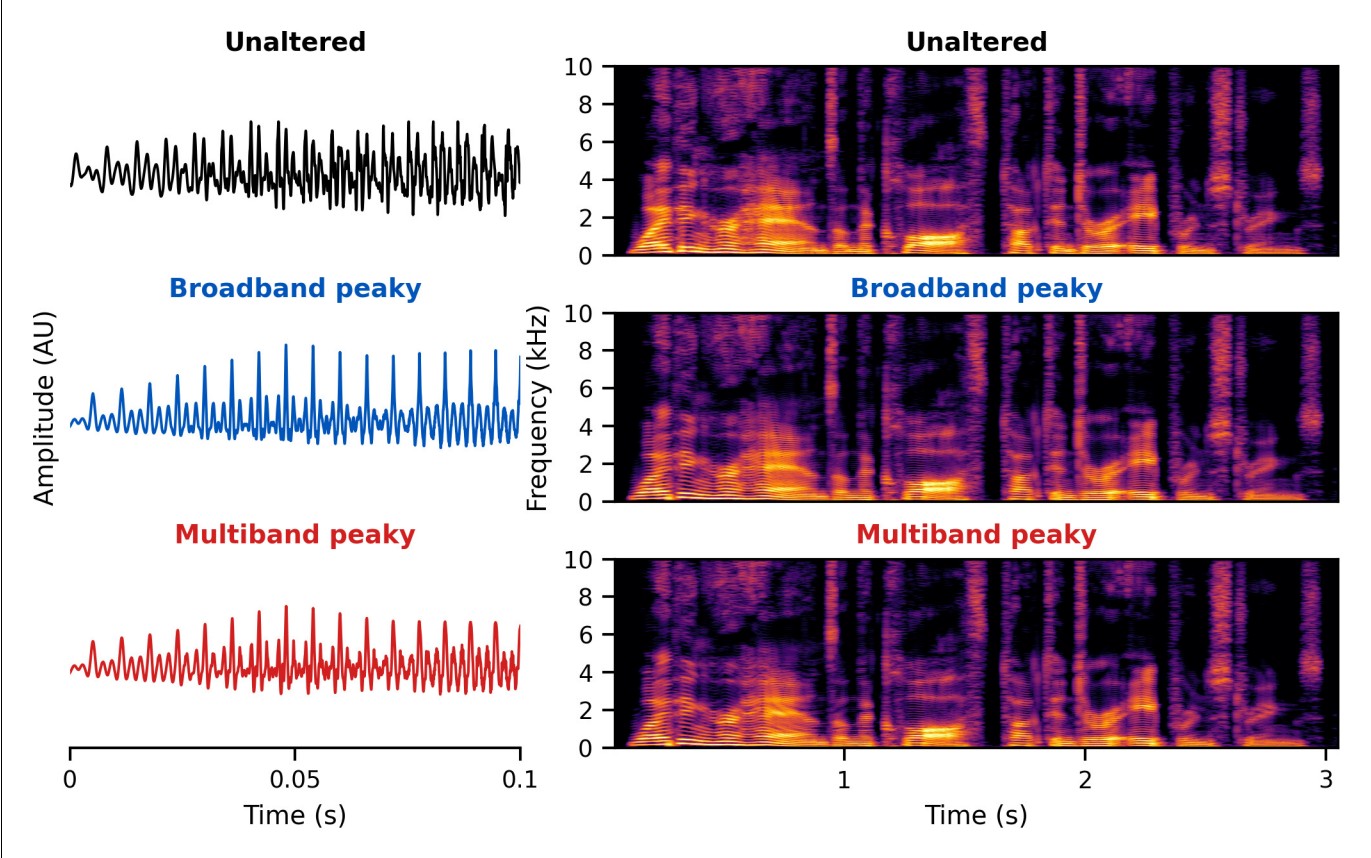

**Figure 13.** Unaltered speech waveform (top left) and spectrogram (top right) compared to re-synthesized broadband peaky speech (middle left and right) and multiband peaky speech (bottom left and right). Comparing waveforms shows that the peaky speech is as 'click-like' as possible, while comparing the spectrograms shows that the overall spectrotemporal content that defines speech is basically unchanged by the re-synthesis. A naïve listener is unlikely to notice that any modification has been performed, and subjective listening confirms the similarity. Yellow/lighter colors represent larger amplitudes than purple/darker colors in the spectrogram. See supplementary files for audio examples of each stimulus type for both narrators.

in creating peaky speech was to use speech processing software (PRAAT, RRID:SCR_016564; *Boersma and Weenink, 2018*) to extract the times of these glottal pulses. Sections of speech where glottal pulses were within 17 ms of each other were considered voiced (vowels and voiced consonants like /z/), as 17 ms is the longest inter-pulse interval one would expect in natural speech because it is the inverse of 60 Hz, the lowest pitch at which someone with a deep voice would likely speak. A longer gap in pulse times was considered a break between voiced sections. These segments were identified in a 'mixer' function of time, with indices of 1 indicating unvoiced and 0 indicating voiced segments (and would later be responsible for time-dependent blending of re-synthesized and natural speech, hence its name). Transitions of the binary mixer function were smoothed using a raised cosine envelope spanning the time between the first and second pulses, as well as the last two pulses of each voiced segment. During voiced segments, the glottal pulses set the fundamental frequency of speech (i.e., pitch), which were allowed to vary from a minimum to maximum of 60–350 Hz for the male narrator and 90–500 Hz for the female narrator. For the male and female narrators, these pulses gave a mean ± SD fundamental frequency (i.e., pulse rate) in voiced segments of 115.1 ± 6.7 Hz and 198.1 ± 20 Hz, respectively, and a mean ± SD pulses per second over the entire 64 s, inclusive of unvoiced periods and silences, of 69.1 ± 5.7 Hz and 110.8 ± 11.4, respectively. These pulse times were smoothed using 10 iterations of replacing pulse time $p_i$ with the mean of pulse times $p_{i-1}$ to $p_{i+1}$ if the $log_2$ absolute difference in the time between $p_i$ and $p_{i-1}$ and $p_{i+1}$ was less than $log_2(1.6)$.

The fundamental frequency of voiced speech is dynamic, but the signal always consists of a set of integer-related frequencies (harmonics) with different amplitudes and phases. To create the

waveform component at the fundamental frequency, $f_0(t)$, we first created a phase function, $\varphi(t)$, which increased smoothly by $2\pi$ between glottal pulses within the voiced sections as a result of cubic interpolation. We then computed the spectrogram of the unaltered speech waveform – which is a way of analyzing sound that shows its amplitude at every time and frequency (*Figure 13*, top right) – that we called $A[t, f_0(t)]$. We then created the fundamental component of the peaky speech waveform as

$$h_0(t) = A[t, f_0(t)]\cos[\varphi(t)].$$

This waveform has an amplitude that changes according to the spectrogram but always peaks at the time of the glottal pulses.

Next the harmonics of the speech were synthesized. The $k$ th harmonic of speech is at a frequency of $(k+1)f_0$ so we synthesized each harmonic waveform as

$$h_k(t) = A[t, (k+1)f_0(t)]\cos[(k+1)\varphi(t)].$$

Each of these harmonic waveforms has multiple peaks per period of the fundamental, but every harmonic also has a peak at exactly the time of the glottal pulse. Because of these coincident peaks, when the harmonics are summed to create the re-synthesized voiced speech, there is always a large peak at the time of the glottal pulse. In other words, the phases of all the harmonics align at each glottal pulse, making the pressure waveform of the speech appear 'peaky' (*Figure 13*, left middle).

The resultant re-synthesized speech contained only the voiced segments of speech and was missing unvoiced sounds like /s/ and /k/. Thus the last step was to mix the re-synthesized voiced segments with the original unvoiced parts. This was done by cross-fading back and forth between the unaltered speech and re-synthesized speech during the unvoiced and voiced segments, respectively, using the binary mixer function created when determining where the voiced segments occurred. We also filtered the peaky speech to an upper limit of 8 kHz and used the unaltered speech above 8 kHz to improve the quality of voiced consonants such as /z/. Filter properties for the broadband peaky speech are further described below in the 'Band filters' subsection.

## Multiband peaky speech

The same principles to generate broadband peaky speech were applied to create stimuli designed to investigate the brainstem's response to different frequency bands that comprise speech. This makes use of the fact that over time speech signals with slightly different $f_0$ are independent, or have (nearly) zero cross-correlation, at the lags for the ABR. To make each frequency band of interest independent, we shifted the fundamental frequency and created a fundamental waveform and its harmonics as

$$h_k(t) = A[t, (k+1)f_0(t)]\cos[(k+1)\varphi(t)],$$

where

$$\varphi(t) = 2\pi \int_0^t (f_0(\tau) + f_\Delta) d\tau,$$

and where $f_\Delta$ is the small shift in fundamental frequency.

In these studies, we increased fundamentals for each frequency band by the square root of each successive prime number and subtracting 1, resulting in a few tenths of a hertz difference between bands. The first, lowest frequency band contained the unshifted $f_0$. Responses to this lowest, unshifted frequency band showed some differences from the common component for latencies >30 ms that were not present in the other, higher frequency bands (*Figure 6*, 0–1 kHz band), suggesting some low-frequency privilege/bias in this response. Therefore, we suggest that the following studies create independent frequency bands by synthesizing a new fundamental for each band. The static shifts described above could be used, but we suggest an alternative method that introduces random dynamic frequency shifts of up to ±1 Hz over the duration of the stimulus. From this random frequency shift, we can compute a dynamic random phase shift, to which we also add a random

starting phase, $\theta_\Delta$, which is drawn from a uniform distribution between 0 and $2\pi$. The phase function from the above set of formulae would be replaced with this random dynamic phase function:

$$\varphi(t) = 2\pi \int_0^t [f_0(\tau) + f_\Delta(\tau)]d\tau + \theta_\Delta$$

Validation data from one subject is provided in *Figure 6—figure supplement 1*. Responses from all four bands show more consistent resemblance to the common component, indicating that this method is effective at reducing stimulus-related bias. However, low-frequency-dependent differences remained, suggesting that there is also unique neural-based low-frequency activity to the speech-evoked responses.

This re-synthesized speech was then band-pass filtered to the frequency band of interest (e.g., from 0 to 1 kHz or 2–4 kHz). This process was repeated for each independent frequency band, the bands were mixed together, and then these re-synthesized voiced parts were mixed with the original unaltered voiceless speech. This peaky speech comprised octave bands with center frequencies of 707, 1414, 2929, and 5656 Hz for experiments 1 and 2, and of 500, 1000, 2000, 4000, and 8000 Hz for experiment 3. Note that for the lowest band the actual center frequency was slightly lower because the filters were set to pass all frequencies below the upper cutoff. Filter properties for these two types of multiband speech are shown in the middle and right panels of Figure 16 and further described below in the 'Band filters' subsection. For the dichotic multiband peaky speech, we created 10 fundamental waveforms – two in each of the five filter bands for the two different ears, making the output audio file stereo (or dichotic). We also filtered this dichotic multiband peaky speech to an upper limit of 11.36 kHz to allow for the highest band to have a center frequency of 8 kHz and octave width. The relative mean-squared magnitude in decibels for components of the multiband peaky speech (four filter bands) and dichotic (audiological) multiband peaky speech (five filter bands) are shown in *Figure 14*.

For peaky speech, the re-synthesized speech waveform was presented during the experiment, but the pulse trains were used as the input stimulus for calculating the response (i.e., the regressor, see Response derivation section below). These pulse trains all began and ended together in conjunction with the onset and offset of voiced sections of the speech. To verify which frequency ranges of the multiband pulse trains were independent across frequency bands and would thus yield truly band-specific responses, we conducted spectral coherence analyses on the pulse trains. All 60

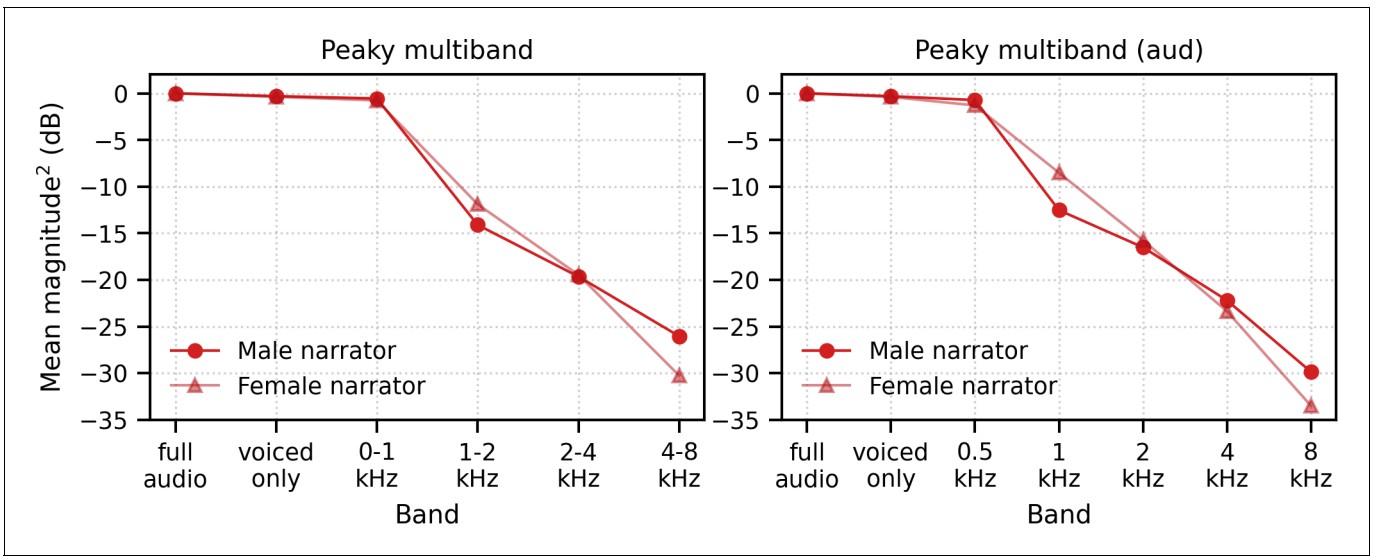

**Figure 14.** Relative mean-squared magnitude in decibels of multiband peaky speech with four filter bands (left) and five filter bands (right) for male- (dark red circles) and female-narrated (light red triangles) speech. The full audio comprises unvoiced and re-synthesized voiced sections, which was presented to the subjects during the experiments. The other bands reflect the relative magnitude of the voiced sections (voiced only), and each filtered frequency band.

unique 64 s sections of each male- and female-narrated multiband peaky speech used in the three experiments were sliced into 1 s segments for a total of 3840 slices. Phase coherence across frequency was then computed across these slices for each combination of pulse trains according to the formula

$$C_{xy} = \frac{|\,E[\mathcal{F}\{x_i\}^*\,\mathcal{F}\{y_i\}]\,|}{\sqrt{E[\mathcal{F}\{x_i\}^*\,\mathcal{F}\{x_i\}]\,E[\mathcal{F}\{y_i\}^*\,\mathcal{F}\{y_i\}]}}$$

where $C_{xy}$ denotes coherence between bands $x$ and $y$, $E[\,]$ the average across slices, $\mathcal{F}$ the fast Fourier transform, * the complex conjugation, $x_i$ the pulse train for slice $i$ in band $x$, and $y_i$ the pulse train for slice $i$ in band $y$.

Spectral coherence for each narrator is shown in *Figure 15*. For the four-band multiband peaky speech used in experiments 1 and 2, there were six pulse train comparisons. For the audiological multiband peaky speech used in experiment 3, there were five bands for each of the two ears, resulting in 10 pulse trains and 45 comparisons. All 45 comparisons are shown in *Figure 15A* for the stimuli created with static frequency shifts, and the six comparisons for the stimuli created with random dynamic frequency shifts (used in the pilot experiment) are shown in *Figure 15B*. Pulse trains were coherent (>0.1) up to a maximum of 71 and 126 Hz for male- and female-narrated speech, respectively, which roughly correspond to the mean ± SD pulse rates (calculated as total pulses/64 s) of 69.1 ± 5.7 Hz and 110.8 ± 11.4, respectively. This means that above ~130 Hz the stimuli were no longer coherent and evoked frequency-specific responses. Importantly, responses would be to correlated stimuli, that is, not frequency-specific, at frequencies below this cutoff and would result in a low-frequency response component that is present in (or common to) all band responses. There were two separated collections of curves for the stimuli with static frequency shifts, which stemmed from the shifted frequency bands having different coherence with each other than the lowest unshifted band. This stimulus-related bias in the lowest frequency band was reduced by using

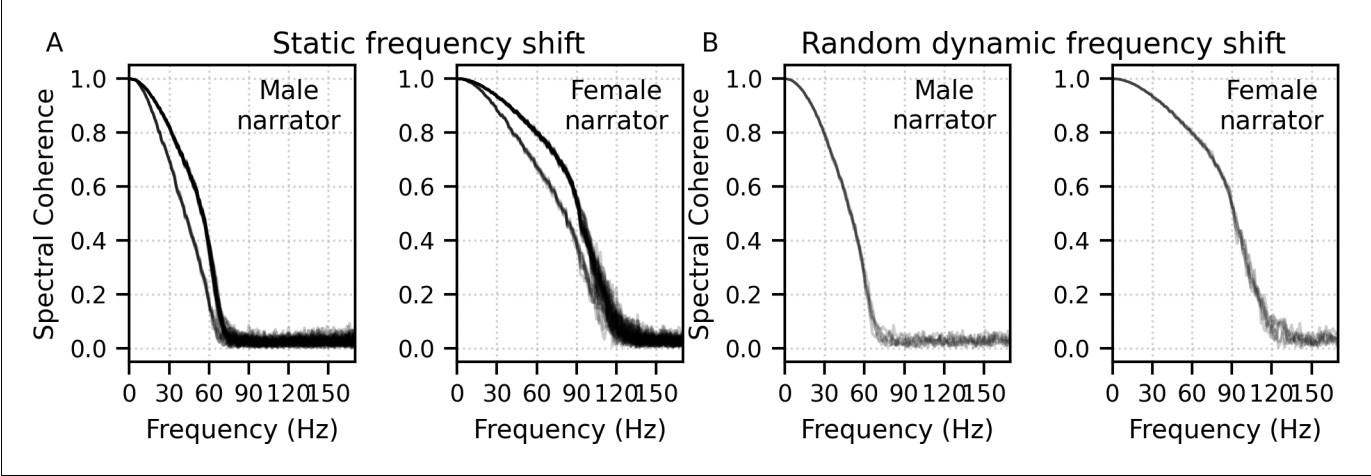

**Figure 15.** Spectral coherence of pulse trains for multiband peaky speech narrated by a male (left) and female (right). Spectral coherence was computed across 1 s slices from 60 unique 64 s multiband peaky speech segments (3840 total slices) for each combination of bands. Each light gray line represents the coherence for one band comparison. (A) There were 45 comparisons across the 10-band (audiological) speech used in experiment 3 (5 frequency bands × 2 ears). The lowest band was unshifted, and the other nine bands had static frequency shifts. (B) There were six comparisons across four pulse trains of the bands in the pilot experiment, which all had dynamic random frequency shifts. Pulse trains (i.e., the input stimuli, or regressors, for the deconvolution) were frequency-dependent (coherent) below 72 Hz for the male multiband speech and 126 Hz for the female multiband speech. The online version of this article includes the following figure supplement(s) for figure 15:

**Figure supplement 1.** Comparison of the common component derived from the average response to six fake pulse trains that were created using static frequency shifts (solid, darker lines; used in the paper) or dynamic random frequency shifts (dashed, lighter lines, pilot data and suggested in 'Materials and methods').

**Figure supplement 2.** Multiband stimuli responses for male (left) and female (right) derived by deconvolving the absolute value of the dichotic (stereo) multiband peaky audio (from experiment 3) with the 10 associated pulse trains – five pulse trains were used for each band in each ear ('correct' pulse trains, top row).

dynamic random frequency shifts (used in the pilot experiment), as indicated by the similar spectral coherence curves shown in *Figure 15B*.

To identify the effect of the low-frequency stimulus coherence in the responses, we computed the common component across pulse trains by creating an averaged response to six additional 'fake' pulses trains that were created during stimulus design but were not used during the creation of the multiband peaky speech wav files. The common component was assessed for both 'fake' pulse trains taken from shifts lower than the original fundamental frequency and those taken from shifts higher than the highest 'true' re-synthesized fundamental frequency. For the dynamic frequency shift method (pilot experiment), an additional six pulse trains were created. *Figure 15—figure supplement 1* shows that the common component was similar for 'fake' pulse trains created using static or dynamic random frequency shifts. To assess frequency-specific responses to multiband speech, we subtracted this common component from the band responses. Alternatively, one could simply high-pass the stimuli at 150 Hz using a first-order causal Butterworth filter (being mindful of edge artifacts). However, this high-pass filtering reduces response amplitude and may affect response detection (see Results for more details).

We also verified the independence of the stimulus bands by treating the regressor pulse train as the input to a system whose output was the rectified stimulus audio and performed deconvolution (see Deconvolution and Response derivation sections below). Further details are provided in *Figure 15—figure supplement 2*. The responses showed that the non-zero responses only occurred when the correct pulse train was paired with the correct audio.

## Band filters

Because the fundamental frequencies for each frequency band were designed to be independent over time, the band filters for the speech were designed to cross over in frequency at half power. To make the filter, the amplitude was set by taking the square root of the specified power at each frequency. Octave band filters were constructed in the frequency domain by applying trapezoids – with center bandwidth and roll-off widths of 0.5 octaves. For the first (lowest frequency) band, all frequencies below the high-pass cutoff were set to 1, and likewise for all frequencies above the low-pass cutoff for the last (highest frequency) band were set to 1 (*Figure 16*, top row). The impulse response of the filters was assessed by shifting the inverse fast Fourier transform (IFFT) of the bands so that time zero was in the center, and then applied a Nuttall window, thereby truncating the impulse response to length of 5 ms (*Figure 16*, middle row). The actual frequency response of the filter bands was assessed by taking the fast Fourier transform (FFT) of the impulse response and plotting the magnitude (*Figure 16*, bottom row).

As mentioned above, broadband peaky speech was filtered to an upper limit of 8 kHz for diotic peaky speech and 11.36 kHz for dichotic peaky speech. This band filter was constructed from the second last octave band filter from the multiband filters (i.e., the 4–8 kHz band from the top-middle of *Figure 16*, dark red line) by setting the amplitude of all frequencies less than the high-pass cutoff frequency to 1 (*Figure 16*, top left, blue line). As mentioned above, unaltered (unvoiced) speech above 8 kHz (diotic) or 11.36 kHz (dichotic) was mixed with the broadband and multiband peaky speech, which was accomplished by applying the last (highest) octave band filter (8+ or 11.36+ kHz band, black line) to the unaltered speech and mixing this band with the re-synthesized speech using the other bands.

## Alternating polarity

To limit stimulus artifact, we also alternated polarity between segments of speech. To identify regions to flip polarity, the envelope of speech was extracted using a first-order causal Butterworth low-pass filter with a cutoff frequency of 6 Hz applied to the absolute value of the waveform. Then, flip indices were identified where the envelope became less than 1% of the median envelope value, and a function that changed back and forth between 1 and −1 at each flip index was created. This function of spikes was smoothed using another first-order causal Butterworth low-pass filter with a cutoff frequency of 10,000 Hz, which was then multiplied with the re-synthesized speech before saving to a wav file.

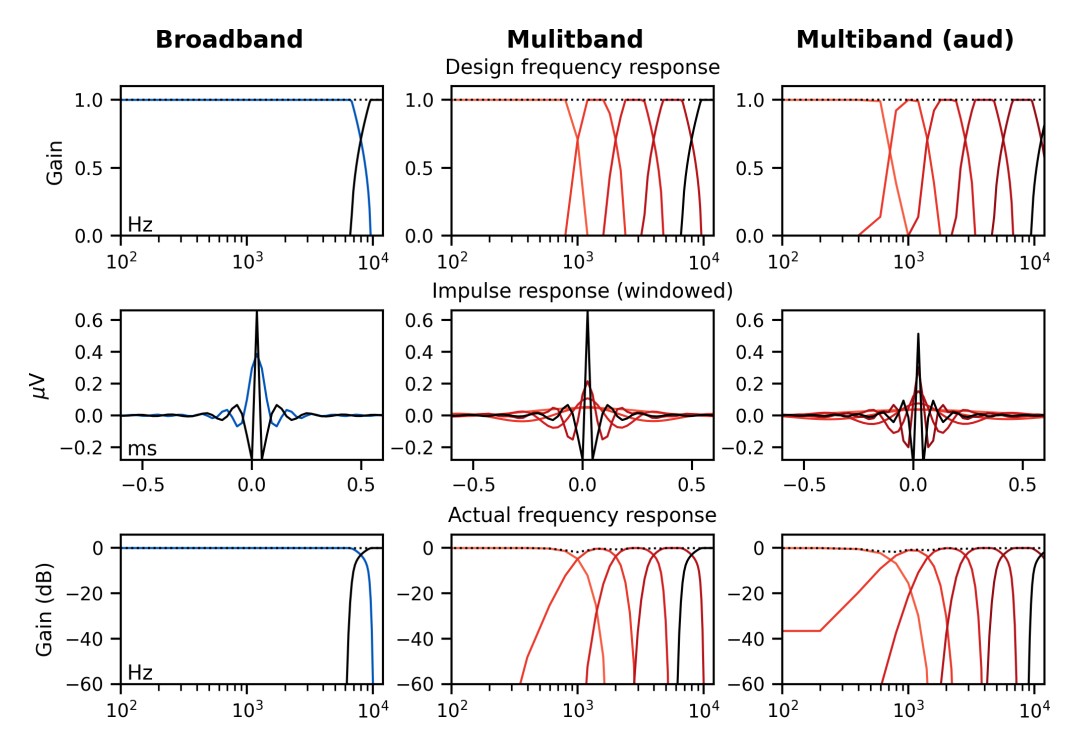

**Figure 16.** Octave band filters used to create re-synthesized broadband peaky speech (left, blue), diotic multiband peaky speech with four bands (middle, red), and dichotic multiband peaky speech using five bands with audiological center frequencies (right, red). The last band (2nd, 5th, and 6th, respectively, black line) was used to filter the high frequencies of unaltered speech during mixing to improve the quality of voiced consonants. The designed frequency response using trapezoids (top) was converted into time-domain using IFFT, shifted, and Nuttall windowed to create impulse responses (middle), which were then used to assess the actual frequency response by converting into the frequency domain using FFT (bottom).

## Response derivation

### Deconvolution

The peaky speech ABR was derived by using deconvolution, as in previous work (*Maddox and Lee, 2018*), though the computation was performed in the frequency domain for efficiency. The speech was considered the input to a linear system whose output was the recorded EEG signal, with the ABR computed as the system's impulse response. As in *Maddox and Lee, 2018*, for the unaltered speech, we used the half-wave rectified audio as the input waveform. Half-wave rectification was accomplished by separately calculating the response to all positive and negative values of the input waveform for each epoch and then combining the responses together during averaging. For our new re-synthesized peaky speech, the input waveform was the sequence of impulses that occurred at the glottal pulse times and corresponded to the peaks in the waveform. *Figure 17* shows a section of stimulus and the corresponding input signal of glottal pulses used in the deconvolution.

The half-wave rectified waveforms and glottal pulse sequences were corrected for the small clock differences between the sound card and EEG system and then downsampled to the EEG sampling frequency prior to deconvolution. To avoid temporal splatter due to standard downsampling, the pulse sequences were resampled by placing unit impulses at sample indices closest to each pulse time. Regularization was not necessary because the amplitude spectra of these regressors were sufficiently broadband. For efficiency, the time domain response waveform, $w$, for a given 64 s epoch was calculated using frequency domain division for the deconvolution, with the numerator the cross-spectral density (corresponding to the cross-correlation in the time domain) of the stimulus regressor and EEG response, and the denominator the power spectral density of the stimulus regressor (corresponding to its autocorrelation in the time domain). For a single epoch, that would be

$$w = \mathcal{F}^{-1}\left\{\frac{\mathcal{F}\{x\}^* \, \mathcal{F}\{y\}}{\mathcal{F}\{x\}^* \, \mathcal{F}\{x\}}\right\}$$

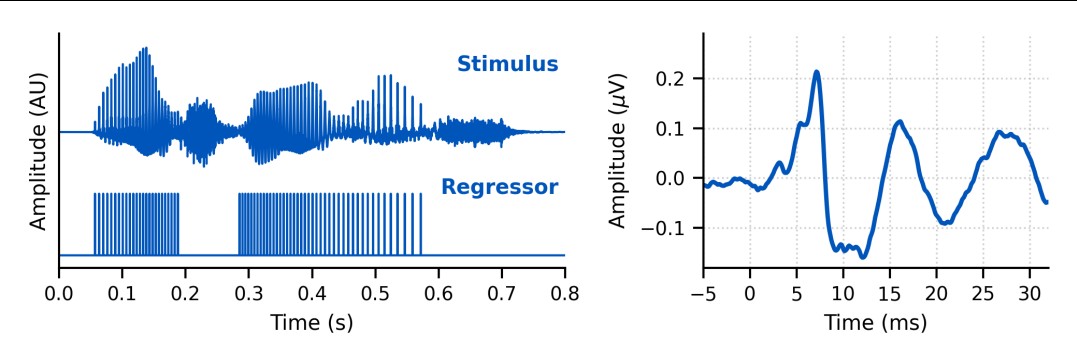

**Figure 17.** Example stimulus, regressor, and deconvolved response. Left: A segment of broadband peaky speech stimulus (top) and the corresponding glottal pulse train (bottom) used in calculating the broadband peaky speech response. Right: An example broadband peaky speech response from a single subject. The response shows auditory brainstem response waves I, III, and V at ~3, 5, and 7 ms, respectively. It also shows later peaks corresponding to thalamic and cortical activity at ~17 and 27 ms, respectively.

where $\mathcal{F}$ denotes the fast Fourier transform, $\mathcal{F}^{-1}$ the inverse fast Fourier transform, * the complex conjugation, $x$ the input stimulus regressor (half-wave rectified waveform or glottal pulse sequence), and $y$ the EEG data for each epoch. We used methods incorporated into the MNE-python package (RRID:SCR_005972; *Gramfort et al., 2013*). In practice, we made adaptations to this formula to improve the SNR with Bayesian-like averaging (see below). For multiband peaky speech, stimuli with slightly different fundamental frequencies will be independent, yielding independent ABRs. Therefore, the same EEG was deconvolved with each band's pulse train to derive the ABR and MLR to each frequency band, and then with an additional six 'fake' pulse trains – pulse trains with slightly different fundamental frequencies that were not used to create the presented multiband peaky speech stimuli – to derive the common component across bands due to the pulse train coherence at low frequencies (shown in *Figure 15*). The averaged response across these six fake pulse trains, or common component, was then subtracted from the multiband responses to identify the frequency-specific band responses.

## Response averaging

The quality of the ABR waveforms as a function of each type of stimulus was of interest, so we calculated the averaged response after each 64 s epoch. We followed a Bayesian-like process (*Elberling and Wahlgreen, 1985*) to account for variations in noise level across the recording time (such as slow drifts or movement artifacts) and avoid rejecting data based on thresholds. Each epoch was weighted by its inverse variance, $1/\sigma_i^2$, to the sum of the inverse variances of all epochs. Thus, epoch weights, $b_i$, were calculated as follows:

$$b_i = \frac{1/\sigma_i^2}{\sum_{i=1}^n 1/\sigma_i^2}$$

where $i$ is the epoch number and $n$ is the number of epochs collected. For efficiency, weighted averaging was completed during deconvolution. Because auto-correlation of the input stimulus (denominator of the frequency domain division) was similar across epochs, it was averaged with equal weighting. Therefore, the numerator of the frequency domain division was summed across weighted epochs and the denominator averaged across epochs, according to the following formula:

$$w = \mathcal{F}^{-1}\left\{\frac{\sum_{i=1}^n b_i \mathcal{F}\{x_i\}^* \mathcal{F}\{y_i\}}{\sum_{i=1}^n \frac{1}{n}\mathcal{F}\{x_i\}^* \mathcal{F}\{x_i\}}\right\}$$

where $w$ is the average response waveform, $i$ is again the epoch number, and $n$ is the number of epochs collected.

Due to the circular nature of the discrete frequency domain deconvolution, the resulting response has an effective time interval of [0, 32] s at the beginning and [−32, 0] s at the end, so that concatenating the two – with the end first – yields the response from [−32, 32] s. Consequently, to avoid edge artifacts, all filtering was performed after the response was shifted to the middle of the 64 s time window. To remove high-frequency noise and some low-frequency noise, the average waveform was band-pass filtered between 30–2000 Hz using a first-order causal Butterworth filter. An example of this weighted average response to broadband peaky speech is shown in *Figure 17* (right). This bandwidth of 30–2000 Hz is sufficient to identify additional waves in the brainstem and MLR (ABR and MLR, respectively). To further identify earlier waves of the ABRs (i.e., waves I and III), responses were high-pass filtered at 150 Hz using a first-order causal Butterworth filter. This filter was determined to provide the best morphology without compromising the response by comparing responses filtered with common high-pass cutoffs of 1, 30, 50, 100, and 150 Hz each combined with first-, second-, and fourth-order causal Butterworth filters.

## Response normalization

An advantage of this method over our previous one (*Maddox and Lee, 2018*) is that because the regressor comprises unit impulses the deconvolved response is given in meaningful units, which are the same as the EEG recording, namely microvolts. With a continuous regressor, like the half-wave rectified speech waveform, this is not the case. Therefore, to compare responses to half-wave rectified speech versus glottal pulses, we calculated a normalization factor, $g$, based on data from all subjects:

$$g = \frac{1/n \sum_{i=1}^{n} \sigma_{u,i}}{1/n \sum_{i=1}^{n} \sigma_{p,i}}$$

where $n$ is the number of subjects, $\sigma_{u,i}$ is the SD of subject $i$'s response to unaltered speech between 0–20 ms, and $\sigma_{p,i}$ is the same for the broadband peaky speech. Each subject's responses to unaltered speech were multiplied by this normalization factor to bring these responses within a comparable amplitude range as those to broadband peaky speech. Consequently, amplitudes were not compared between responses to unaltered and peaky speech. This was not our prime interest, rather we were interested in latency and presence of canonical component waves. In this study, the normalization factor was 0.26, which cannot be applied to other studies because this number also depends on the scale when storing the digital audio. In our study, this unitless scale was based on a root-mean-square amplitude of 0.01. The same normalization factor was used when the half-wave rectified speech was used as the regressor with EEG collected in response to unaltered speech, broadband peaky speech, and multiband peaky speech (*Figure 2*, *Figure 4*, *Figure 4—figure supplement 1*).

## Response SNR calculation

We were also interested in the recording time required to obtain robust responses to re-synthesized peaky speech. Therefore, we calculated the time it took for the ABR and MLR to reach a 0 dB SNR. We chose 0 dB SNR based on visual assessment of when waveforms were easily inspected and based on what we have done previously (*Maddox and Lee, 2018*; *Polonenko and Maddox, 2019*). The SNR of each waveform in dB, $SNR_w$, was estimated as

$$SNR_w = 10 \log_{10} \left[ \frac{\sigma_{S+N}^2 - \sigma_N^2}{\sigma_N^2} \right],$$

where $\sigma_{S+N}^2$ represents the variance (i.e., mean-subtracted energy) of the waveform between 0 and 15 ms or 30 ms for the ABR and MLR, respectively (contains both component signals and noise, $S+N$), and $\sigma_N^2$ represents the variance of the noise, $N$, estimated by averaging the variances of 15 ms (ABR) to 30 ms (MLR) segments of the pre-stimulus baseline between −480 and −20 ms. Then, the SNR for 1 min of recording, $SNR_{60}$, was computed from the $SNR_w$ as

$$SNR_{60} = SNR_w + 10 \log_{10}[60/t_w],$$

where $t_w$ is the duration of the recording in seconds, as specified in the 'Speech stimuli and

conditions' section. For example, in experiment 3, the average waveform resulted from 64 min of recording, or a $t_w$ of 3840 s. The time to reach 0 dB SNR for each subject, $t_{0dB\ SNR}$, was estimated from this $SNR_{60}$ by

$$t_{0dB\ SNR} = 60 \times 10^{-SNR_{60}/10}.$$

Cumulative density functions were used to show the proportion of subjects that reached an SNR $\geq$ 0 dB and determine the necessary acquisition times that can be expected for each stimulus on a group level.

## Statistical analyses

### Waveform morphology

Data were checked for normality using the Shapiro–Wilk test. Waveform morphology of responses to different narrators was compared using Pearson correlations of the responses between 0 and 15 ms for the ABR waveforms or 0 and 40 ms for both ABR and MLR waveforms. The Wilcoxon signed-rank test was used to determine whether narrator differences (waveform correlations) were significantly different than the correlations of the same EEG split into even and odd epochs with equal numbers of epochs from each narrator. For experiment 1, responses containing an equal number of epochs from the first and second half of the recording had a median (interquartile range) correlation coefficient of 0.96 (0.95–0.98) for the 0–15 ms lags, indicating good replication.

### Peak latencies

The intraclass correlation coefficient type 3 (absolute agreement) was used to verify good agreement in peak latencies chosen by an experienced audiologist and neuroscientist (MJP) at two different time points, 3 months apart. The ICC3 for all peaks were $\geq$0.90, indicating excellent reliability for the chosen peak latencies.

Independent t-tests with $\mu$ = 0 were conducted on the peak latency differences of ABR/MLR waves for unaltered and broadband peaky speech. For multiband peaky speech, the change in peak latency, $\tau$, with frequency band was modeled using a power law regression (*Harte et al., 2009*; *Neely et al., 1988*; *Rasetshwane et al., 2013*; *Strelcyk et al., 2009*) according to the formula

$$\tau(f) = a + bf^{-d}$$

where

$$a = \tau_{synaptic} + \tau_{I-V},$$

and where $f$ is the band center frequency normalized to 1 kHz (i.e., divided by 1000), $\tau_{synaptic}$ is the synaptic delay (assumed to be 0.8 ms; *Eggermont, 1979*; *Strelcyk et al., 2009*), and $\tau_{I-V}$ is the I–V inter-wave delay from the subjects' responses to broadband peaky speech. The power law model was completed for each filter cutoff of 30 and 150 Hz in the log–log domain, according to the formula

$$\log_{10}(\tau(f) - a) = \log_{10} b + (-d)\log_{10} f$$

and using linear mixed effects regression to estimate the terms $d$ and $b$ (calculated as $b$ = 10$^{intercept}$). For subjects who did not have an identifiable wave I in the broadband peaky response, the group mean I–V delay was used – this occurred for 1 of 22 subjects in experiment 1 and 2 of 11 subjects for responses to the female narrator in experiment 2. Because only multiband peaky speech was presented in experiment 3, the mean I–V intervals from experiment 2 were used for each subject for experiment 3. Random effects of subject and each frequency band term were included to account for individual variability that is not generalizable to the fixed effects. The mixed effects regressions were performed using the lme4 (RRID:SCR_015654) and lmerTest (RRID:SCR_015656) packages in R (RRID:SCR_001905) and RStudio (RRID:SCR_000432) (*Bates et al., 2015*; *Kuznetsova et al., 2017*; *R Development Core Team, 2020*). A power analysis was completed using the simR package (RRID: SCR_019287; *Green and MacLeod, 2016*), which uses a likelihood ratio test on 1000 Monte Carlo permutations of the response variables based on the fitted model.

## Data availability

Python code is available on the lab GitHub account (https://github.com/maddoxlab/peaky-speech; *Polonenko, 2021*; copy archived at swh:1:rev:3c1e5b62d5bb3a5b9cc6130cb3db651bc73-b3ecd). All EEG recordings are available in the EEG-BIDS format (*Pernet et al., 2019*) on Dryad (https://doi.org/10.5061/dryad.12jm63xwd). Stimulus files and python code necessary to derive the peaky speech responses are deposited to the same Dryad repository.

## Acknowledgements

We thank Sara Fiscella for assistance with recruitment. We would also like to thank the reviewers, whose suggestions significantly strengthened the manuscript. This work was supported by the National Institute for Deafness and Other Communication Disorders (R00DC014288) awarded to RKM.

## Additional information

### Funding

| Funder | Grant reference number | Author |
| --- | --- | --- |
| National Institute on Deafness and Other Communication Disorders | R00DC014288 | Ross K Maddox |

The funders had no role in study design, data collection and interpretation, or the decision to submit the work for publication.

### Author contributions

Melissa J Polonenko, Conceptualization, Data curation, Software, Formal analysis, Supervision, Validation, Investigation, Visualization, Methodology, Writing - original draft, Project administration, Writing - review and editing; Ross K Maddox, Conceptualization, Resources, Software, Formal analysis, Supervision, Funding acquisition, Validation, Investigation, Visualization, Methodology, Writing - original draft, Project administration, Writing - review and editing

### Author ORCIDs

Melissa J Polonenko (iD) https://orcid.org/0000-0003-1914-6117
Ross K Maddox (iD) https://orcid.org/0000-0003-2668-0238

### Ethics

Human subjects: All subjects gave written informed consent before the experiment began. All experimental procedures were approved by the University of Rochester Research Subjects Review Board. (#1227).

### Decision letter and Author response

Decision letter https://doi.org/10.7554/eLife.62329.sa1
Author response https://doi.org/10.7554/eLife.62329.sa2

## Additional files

### Supplementary files

• Supplementary file 1. Details of statistical models. (**A**) LMER model formula and summary output for multiband peaky speech in experiment 1. (**B**) LMER model formula and summary output for multiband peaky speech in experiment 2. (**C**) LMER model formula and summary output for multiband peaky speech in experiment 3.

• Audio file 1. Unaltered speech sample from the male narrator (*The Alchemyst*; *Scott, 2007*).

- Audio file 2. Broadband peaky speech sample from the male narrator (*The Alchemyst*; *Scott, 2007*).
- Audio file 3. Multiband peaky speech sample from the male narrator (*The Alchemyst*; *Scott, 2007*).
- Audio file 4. Unaltered speech sample from the female narrator (*A Wrinkle in Time*; *L'Engle, 2012*).
- Audio file 5. Broadband peaky speech sample from the female narrator (*A Wrinkle in Time*; *L'Engle, 2012*).
- Audio file 6. Multiband peaky speech sample from the female narrator (*A Wrinkle in Time*; *L'Engle, 2012*).
- Transparent reporting form

### Data availability

Python code is available on the lab GitHub account (https://github.com/maddoxlab/peaky-speech; copy archived at https://archive.softwareheritage.org/swh:1:rev:3c1e5b62d5b-b3a5b9cc6130cb3db651bc73b3ecd/). All EEG recordings are posted in the EEG-BIDS format (Pernet et al., 2019) to Dryad (https://doi.org/10.5061/dryad.12jm63xwd). Stimulus files necessary to derive the peaky speech responses are deposited in the same Dryad repository.

The following dataset was generated:

| Author(s) | Year | Dataset title | Dataset URL | Database and Identifier |
|---|---|---|---|---|
| Polonenko MJ, Maddox RK | 2020 | Exposing distinct subcortical components of the auditory brainstem response evoked by continuous naturalistic speech | http://dx.doi.org/10.5061/dryad.12jm63xwd | Dryad Digital Repository, 10.5061/dryad.12jm63xwd |

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
