## [Decision Letter]

**Acceptance summary:**

This manuscript describes an alteration to speech to make it more peaky, The authors show that the developed peaky speech, in contrast to naturalistic speech, allows measurements of distinct subcortical components of the neural response. This development may allow further and more refined investigations of the contribution of different subcortical structures to speech processing as well as to hearing deficits.

**Decision letter after peer review:**

Thank you for submitting your article "Exposing distinct subcortical components of the auditory brainstem response evoked by continuous naturalistic speech" for consideration by *eLife*. Your article has been reviewed by three peer reviewers, including Tobias Reichenbach as the Reviewing Editor and Reviewer #1, and the evaluation has been overseen by Andrew King as the Senior Editor. The following individual involved in review of your submission has agreed to reveal their identity: Jonathan Z Simon (Reviewer #3).

The reviewers have discussed the reviews with one another and the Reviewing Editor has drafted this decision to help you prepare a revised submission.

Summary:

This manuscript describes a type of alteration to speech to make it more peaky, with the goal of inducing stronger responses in the auditory brainstem. Recent work has employed naturalistic speech to investigate subcortical mechanisms of speech processing. However, previous methods were ill equipped to tease apart the neural responses in different parts of the brainstem. The authors show that their speech manipulation improves this: the peaky speech that they develop allows segregation of different waves of the brainstem response. This development may allow further and more refined investigations of the contribution of different parts of the brainstem to speech processing, as well as to hearing deficits.

Essential revisions:

– Despite repeated claims, we are not sure that a convincing case is made here that this method can indeed provide insight into how speech is processed in early auditory pathway. The response is essentially a click-like response elicited by the glottal pulses in the stimulus; it averages out information related to dynamic variations in envelope and pitch that are essential for speech perception; at the same time, it is highly sensitive to sound features that do not affect speech perception. What reason is there to assume that these responses contain information that is specific or informative about speech processing?

– Similarly, the claim that the methodology can be used as a clinical application is not convincing. It is not made clear what pathology these responses can detect that current ABR methods cannot, or why. As explained in the Discussion, the response size is inherently smaller than standard ABRs because of the higher repetition rate of the glottal pulses, and the response may depend on more complex neural interactions that would be difficult to quantify. Do these features not make them less suitable for clinical use?

– It needs to be rigorously confirmed that the earliest responses are not contaminated or influenced by responses from later sources. There seems to be some coherent activity or offset in the baseline (pre 0 ms), in particular with the lower filter cut off. One way to test this might be to simulate a simple response by filtering and time shifting the stimulus waveforms, adding these up plus realistic noise, and applying the deconvolution to see whether the input is accurately reproduced. It might be useful to see how the response latencies and amplitudes correlate to those of conventional click responses, and how they depend on stimulus level.

– The multiband responses show a variation of latency with frequency band that indicates a degree of cochlear frequency specificity. The latency functions reported here looks similar to those obtained by Don et al. 1993 for derived band click responses, but the actual numbers for the frequency dependent delays (as estimated by eye from Figures 4,6 and 7) seem shorter than those reported for wave V at 65 dB SPL (Don et al. 1993 table II). The latency function would be better fitted to an exponential, as in Strelcyk et al. 2009 (Equation 1), than a quadratic function; the fitted exponent could be directly compared to their reported value.

– The fact that differences between narrators leads to changes to the ABR response is to be expected, and was already reported in Maddox and Lee, 2018. We are not sure why this issue needs to be examined and discussed at such length here. The space devoted to discussing the recording time also seems very long. Neither Abstract or Introduction refers to these topics, and they seem to be side-issues that could be summarised and discussed much more briefly.

– The authors motivate the work from the use of naturalistic speech, and the application of the developed method to investigate, for instance, speech-in-noise deficits. But they do not discuss how comprehensible the peaky speech in fact is. We would therefore like to see behavioural experiments that quantitatively compare speech-in-noise comprehension, for example SRTs, for the unaltered speech and the peaky speech. Without such a quantification, it is impossible to fully judge the usefulness of the reported method for further research and clinical applications.

– The neural responses to unaltered speech and to peaky speech are analysed by two different methods. For unaltered speech, the authors use the half-wave rectified waveform as the regressor. For peaky speech, however, the regressor is a series of spikes that are located at the timings of the glottal pulses. Due to this rather different analysis, it is impossible to know to which degree the differences in the neural responses to the two types of speech that the authors report are due to the different speech types, or due to the different analysis techniques. The authors should therefore use the same analysis technique for both types of speech. It might be most sensible to analyse the unaltered speech through a regressor with spikes at the glottal pulses a well. In addition, it would be good to see a comparison, say of a SNR, when the peaky speech is analysed through the half-wave rectified waveform and through the series of spikes. This would also further motivate the usage of the regressor with the series of spikes.

– Subsection “Frequency-specific responses show frequency-specific lags”. What causes the difference between the effect of high-pass filtering and subtracting the common response? If they serve the same purpose, but have different results, this raises the question which is more appropriate.

– Subsection “Frequency-specific responses show frequency-specific lags” paragraph four. This seems a misinterpretation. The similarity between broadband and summated multiband responses indicates that the band filtered components in the multiband stimulus elicited responses that add linearly in the broadband response. It does not imply that the responses to the different bands originate from non-overlapping cochlear frequency regions.

– Is this measure of SNR appropriate, when the baseline is artificially constructed by deconvolution and filtering? Perhaps noise level could be assessed by applying the deconvolution to a silent recording instead? It might also be useful to have a measure of the replicability of the response.

– "wave III was clearly identifiable in 16 of the 22 subjects": Figure 1 indicates that the word "clearly" may be somewhat generous. It would be worthwhile to discuss wave III and its identifiability in more detail (perhaps its identifiability/non-universality could be compared with that of another less prominent peak in traditionally obtained ABRs?).

[Editors' note: further revisions were suggested prior to acceptance, as described below.]

Thank you for submitting your article "Exposing distinct subcortical components of the auditory brainstem response evoked by continuous naturalistic speech" for consideration by *eLife*. Your article has been reviewed by three peer reviewers, including Tobias Reichenbach as the Reviewing Editor and Reviewer #1, and the evaluation has been overseen by Andrew King as the Senior Editor. The following individual involved in review of your submission has agreed to reveal their identity: Jonathan Z Simon (Reviewer #3).

The reviewers have discussed the reviews with one another and the Reviewing Editor has drafted this decision to help you prepare a revised submission.

This manuscript presents an innovative alteration to speech to make it more peaky, resulting in stronger and better delineated responses in the auditory brainstem. Recent work has indeed employed naturalistic speech to investigate subcortical mechanisms of speech processing. However, previous methods were ill equipped to tease apart the neural responses in different parts of the brainstem. The authors show that their speech manipulation improves this issue: the peaky speech that they develop allows them to segregate different waves of the brainstem response. This development allows further and more refined investigations of the contribution of different parts of the brainstem to speech processing, as well as to hearing deficits.

The authors have made substantial additions and changes in the revised manuscript to complete and improve the analysis. These are overall satisfactory, but we still have some remaining points that we would like the authors to address.

– Due to the additions, the Results section is now even longer, as none of the original text has been removed. We suggest to shorten it by moving some technical details from the Results section to the Materials and methods. For instance, this could concern the details of the peak picking, recording time, waveform repeatability (SNR analysis is implicit in the Recording time section) and details of statistical models. Furthermore, some sentences in the Results seem to serve only to summarize the previous section, which seems unnecessary, or make a general statement that is more suitable for the Discussion.

– "unless presented monaurally by headphones". So these studies did show detectable wave I/III waves when using monaural headphone presentation? This should be more clearly phrased. It seems strange to make this presentation mode sound like a limitation – monaural presentation is standard in clinical audiology. This point raises further questions that had not previously occurred to us: can you clarify whether the peaky speech has also been tested using monaural presentation, and if so how the responses differ from diotic/dichotic presentation (and if not, why was diotic presentation chosen)? How would you confirm that dichotic presentation of the peaky speech produces purely ear-specific responses? Could there be potential binaural interactions or contralateral suppression mechanisms with diotic or dichotic presentation? These questions are relevant and should be briefly addressed.

– We are not sure what "region-specific" means here – please clarify.

– It is more common and reliable to use different peak pickers, rather than the same twice. It probably won't matter much here, but it would be preferable in future analyses.

– It should be explained why the comparison is shown only for the high-pass filter at 30 Hz. The high-pass filter at 150 Hz seems more interesting, as it shows the wave I and III peaks in the peaky speech response that the unaltered speech does not.

– "completed two models". It might be more accurate to write "performed two simulations".

– Subsection “More components of the ABR and MLR are present with broadband peaky than unaltered speech” paragraph three: Does "trial" here mean the time period starting at a glottal pulse? How long did this time period extend from there? From what time point on was the ABR zero-padded? These details should be clarified. It could be informative to also show the input kernels in Figure 3, i.e. the simulated "ABR" that did not show the pre-stimulus component.

– Cortical responses are much larger and likely to be more affected by attention than the much smaller earliest responses. Is it known whether a high-pass filter at 150 Hz is sufficient to reliably remove all cortical contributions within the amplitude range of the ABR wave I? Can a reference be provided for this?

– Strictly speaking, the stimuli differed not only in the narrator's identity but also in text. It would be interesting to comment on whether different texts with the same narrator are likely to give different responses.

– Figure 3 and text related to the more aggressive high-pass filtering at 200 Hz:

With this more aggressive filter, the 8 ms peak has flipped sign and gained additional latency. This needs to be explicitly acknowledged and addressed. It might be due to the peak of that filter's non-linear group delay at its cutoff frequency, combined with an abundance of signal power at the cutoff frequency. But there could be other reasons as well.

– A number of parameter estimates from the power law regression results are written as dimensionless but actually have units of ms. This includes τ_synaptic_ (should be 0.8 ms in several locations), τ_I-V_(in several locations), and a (also in several locations).

---

## [Author Response]

Essential revisions:– Despite repeated claims, we are not sure that a convincing case is made here that this method can indeed provide insight into how speech is processed in early auditory pathway. The response is essentially a click-like response elicited by the glottal pulses in the stimulus; it averages out information related to dynamic variations in envelope and pitch that are essential for speech perception; at the same time, it is highly sensitive to sound features that do not affect speech perception. What reason is there to assume that these responses contain information that is specific or informative about speech processing?

The fact that the peaky speech response resembles the click-evoked ABR is its primary strength—not a weakness. Clicks provide high-fidelity waveforms that can be tied to distinct subcortical sources. Speech stimuli have been used by Kraus and many others to relate neural encoding of natural stimuli to cognition, learning, and intellectual and developmental disorders—but the response components are not easily tied to specific neural sources. Here we offer a tool that allows a subject or patient to listen to long-form natural speech, while acquiring a response that is as interpretable as the click ABR. We believe this strength was not well articulated in our first submission, and have made several changes. We have also emphasized the importance of the spectrotemporal context (namely, speech) in which these responses are acquired, and have been more careful to discuss encoding in addition to processing, while making a case for how this transient-evoked response will indeed allow us to study the latter. Our changes are summarized below:

Added paragraph to Introduction: “Although click responses can assess sound encoding (of clicks) at early stages of the auditory system, a speech-evoked response with the same components would assess region-specific encoding within the acoustical context of the dynamic spectrotemporal characteristics of speech – information that is not possible to obtain from click responses. Furthermore, changes to the amplitudes and latencies of these early components could inform our understanding of speech processing if deployed in experiments that compare conditions requiring different states of processing, such as attended/unattended speech or understood/foreign language. For example, if a wave I from the auditory nerve differed between speech stimuli that were attended versus unattended, then this would add to our current understanding of the brainstem’s role in speech processing. Therefore, a click-like response that is evoked by speech stimuli facilitates new investigations into speech encoding and processing.”

“…Although clicks also evoke responses with multiple component waves, current studies of synaptopathy show quite varied results with clicks and poor correlation with speech in noise (Bramhall et al., 2019; Prendergast et al., 2017). Obtaining click-like responses to stimuli with all of speech’s spectrotemporal richness may provide a better connection to the specific neural underpinnings of speech encoding, similar to how the complex cross-correlation to a fundamental waveform can change based on the context of attention (Forte et al., 2017; Saiz-Alía et al., 2019).”

“Multiband peaky speech will not replace the current frequency specific ABR, but there are situations where it may be advantageous to use speech over tone pips. Measuring waves I, III, and V of high frequency responses in the context of all the dynamics of speech may have applications to studying effects of cochlear synaptopathy on speech comprehension (Bharadwaj et al., 2014; Liberman et al., 2016).”

“Auditory prostheses have algorithms specifically tuned for the spectrotemporal dynamics of speech that behave very differently in response to standard diagnostic stimuli such as trains of clicks or tone pips. Peaky speech responses could allow us to assess how the auditory system is encoding the amplified speech and validate audibility of the hearing aid fittings before the infant or toddler is old enough to provide reliable speech perception testing. Therefore, the ability of peaky speech to yield both canonical waveforms and frequency-specific responses makes this paradigm a flexible method that assesses speech encoding in new ways.”

“…Indeed, previous methods have been quite successful in elucidating cortical processing of speech under these conditions (O’Sullivan et al., 2019; Teoh and Lalor, 2019). […] Finally, the ability to customize peaky speech for measuring frequency-specific responses provides potential applications to clinical research in the context of facilitating assessment of supra-threshold hearing function and changes to how speech may be encoded following intervention strategies and technologies while using a speech stimulus that algorithms in hearing aids and cochlear implants are designed to process.”

– Similarly, the claim that the methodology can be used as a clinical application is not convincing. It is not made clear what pathology these responses can detect that current ABR methods cannot, or why. As explained in the Discussion, the response size is inherently smaller than standard ABRs because of the higher repetition rate of the glottal pulses, and the response may depend on more complex neural interactions that would be difficult to quantify. Do these features not make them less suitable for clinical use?

We have tempered the text and added more text and specific examples to make it clearer why there could be clinical applications. Even though the responses are smaller, the SNR is appropriate, and the responses reflect the context of rates experienced for salient, more naturally experienced stimuli rather than trains of clicks and pips (to which we don’t normally listen).

We do not claim or aim to replace the click and tone pip ABR, but propose ways that these peaky speech responses can add value to clinical applications. The ABR is an index of subcortical activity, and its clinical utility is not confined to diagnosing pathology through a click ABR, or hearing thresholds through the tone pip ABR. For example, we currently do not have a way of measuring how an infant is receiving speech through an auditory prosthesis (beyond parental observation until the infant is old enough to reliably respond) because hearing aids process clicks and tone pips (of traditional ABRs) in a very different way than speech.

The added text:

In the Discussion: “…peaky speech evoked responses with canonical morphology comprised of waves I, III, V, P0, Na, Pa (Figure 1), reflecting neural activity from distinct stages of the auditory system from the auditory nerve to thalamus and primary auditory cortex (e.g., Picton et al., 1974). Although clicks also evoke responses with multiple component waves, current studies of synaptopathy show quite varied results with clicks and poor correlation with speech in noise (Bramhall et al., 2019; Prendergast et al., 2017). Obtaining click-like responses to stimuli with all of speech’s spectrotemporal richness may provide a better connection to the specific neural underpinnings of speech encoding, similar to how the complex cross-correlation to a fundamental waveform can change based on the context of attention (Forte et al., 2017; Saiz-Alía et al., 2019).”

In the Discussion: “… Also, canonical waveforms were derived in the higher frequency bands of diotically presented speech, with waves I and III identifiable in most subjects. […] Therefore, the ability of peaky speech to yield both canonical waveforms and frequency-specific responses makes this paradigm a flexible method that assesses speech encoding in new ways.”

In the Discussion: “Finally, the ability to customize peaky speech for measuring frequency-specific responses provides potential applications to clinical research in the context of facilitating assessment of supra-threshold hearing function and changes to how speech may be encoded following intervention strategies and technologies while using a speech stimulus that algorithms in hearing aids and cochlear implants are designed to process.”

– It needs to be rigorously confirmed that the earliest responses are not contaminated or influenced by responses from later sources. There seems to be some coherent activity or offset in the baseline (pre 0 ms), in particular with the lower filter cut off. One way to test this might be to simulate a simple response by filtering and time shifting the stimulus waveforms, adding these up plus realistic noise, and applying the deconvolution to see whether the input is accurately reproduced. It might be useful to see how the response latencies and amplitudes correlate to those of conventional click responses, and how they depend on stimulus level.

Thank you for this very important suggestion. We performed two models to better understand the pre/peri-stimulus component. A linear model (relating rectified speech to EEG), as suggested, and an additional physiological model of the periphery. We have added a paragraph and figure to the Results section. In a follow-up paper we will investigate how the response latencies and amplitudes correlate with those of the conventional click responses, and how they depend on stimulus level. We believe this is important to investigate as a follow-up but we worry that adding even more analyses or figures could bog down the paper.

In sum, what we found is that there was indeed some spreading of post-stimulus responses into the pre-stimulus period. This spread component was broader than the response components themselves, and so could be removed with more aggressive high-pass filtering when the experiment calls for it. We appreciate the reviewer’s observation and insightful suggestion for how to address it.

The added paragraph and figure with the modeling work is as follows:

“The response to broadband peaky speech showed a small but consistent response at negative and early positive lags (i.e., pre-stimulus) when using the pulse train as a regressor in the deconvolution, particularly when using the lower high-pass filter cutoff of 30 Hz (Figure 2A) compared to 150 Hz (Figure 1). […] Therefore, when doing an experiment where the analysis needs to evaluate specific contributions to the earliest ABR components, we recommend high-pass filtering to help mitigate the complex and time-varying nonlinearities inherent in the auditory system, as well as potential influences by responses from later sources.”

We added this to the filtering considerations in the Discussion:

“When evaluating specific contributions to the earliest waves, we recommend at least a first order 150 Hz high-pass filter or a more aggressive second order 200 Hz high-pass filter to deal with artifacts arising from nonlinearities that are not taken into account by the pulse train regressor or any potential influences by responses from later sources (Figure 3).”

– The multiband responses show a variation of latency with frequency band that indicates a degree of cochlear frequency specificity. The latency functions reported here looks similar to those obtained by Don et al. 1993 for derived band click responses, but the actual numbers for the frequency dependent delays (as estimated by eye from Figures 4,6 and 7) seem shorter than those reported for wave V at 65 dB SPL (Don et al. 1993 table II). The latency function would be better fitted to an exponential, as in Strelcyk et al., 2009 (Equation 1), than a quadratic function; the fitted exponent could be directly compared to their reported value.

Thank you for this suggestion. We re-fit the data with the power law regression performed in the log-log domain using linear mixed effects regression. The Results sections were updated, as well as the supplementary file and statistical analysis section. We updated the Discussion to directly compare our model parameters for wave V to previously reported values. Note: formatting for the equations did not carry over to this document.

“The nonlinear change in peak latency, *τ*, with frequency band was modeled using power law regression according to the formula (Harte et al., 2009; Neely et al., 1988; Rasetshwane et al., 2013; Strelcyk et al., 2009): *τ(f) = a + bf^-d^* where *a = τ_synaptic_+ τ_(I-V)_*, and where *f* is the band center frequency normalized to 1 kHz (i.e., divided by 1000), *τ_synaptic_* is the synaptic delay (assumed to be 0.8; Eggermont, 1979; Strelcyk et al., 2009), and *τ_(I-V)_* is the I-V inter-wave delay from the subjects' responses to broadband peaky speech. […] The significant decrease in latency with frequency band (linear term, slope, *d* : p < 0.001 for 30 and 150 Hz) was shallower (i.e., less negative) for MLR waves compared to the ABR wave V (all p < 0.001 for interactions between wave and the linear frequency band term for 30 and 150 Hz).”

“Therefore, a power law model (see previous subsection for the formula) was completed for waves V, P0, Na, and Pa of responses in the 4 frequency bands that were high-pass filtered at 30 Hz. […] This change with frequency was smaller (i.e., shallower slope) for each MLR wave compared to wave V (p < 0.001 for all interactions between wave and the term for the frequency band). There was a main effect of narrator on peak latencies but no interaction with wave (narrator p = 0.001, wave-narrator interactions p > 0.087).”

“The nonlinear change in wave V latency with frequency was modeled by a power law using log-log mixed effects regression with fixed effects of narrator, ear, logged band center frequency normalized to 1 kHz, and the interactions between narrator and frequency. Random effects included an intercept and frequency term for each subject. The experiment 2 average estimates for *a* of 5.06 and 5.58 for the male- and female-narrated responses were used for each subject. Details of the model are described in Supplementary file 1C. For wave V, the estimated mean parameters were *b* = 4.13 ms (calculated as *b = 10^intercept^*) and *d* = -0.36 for the male narrator, and *b* = 4.25 ms and *d* = -0.41 for the female narrator, which corresponded to previously reported ranges for tone pips and derived-bands at 65 dB ppeSPL (Neely et al., 1988; Rasetshwane et al., 2013; Strelcyk et al., 2009). Latency decreased with increasing frequency (slope, *d*, *p* < 0.001) but did not differ between ears (*p* = 0.265). The *b* parameter differed by narrator but the slope did not change with narrator (interaction with intercept *p* = 0.004, interaction with slope *p* = 0.085).”

In the Discussion: “Peak wave latencies of these responses decreased with increasing band frequency in a similar way to responses evoked by tone pips and derived-bands from clicks in noise (Gorga et al., 1988; Neely et al., 1988; Rasetshwane et al., 2013; Strelcyk et al., 2009), thereby representing activity evoked from different areas across the cochlea. In fact, our estimates of the power law parameters of *a* (the central conduction time), *d* (the frequency dependence) and *b* (the latency corresponding to 1 kHz and 65 dB SPL) for wave V fell within the corresponding ranges that were previously reported for tone pips and derived-bands at 65 dB ppeSPL (Neely et al., 1988; Rasetshwane et al., 2013; Strelcyk et al., 2009).”

In the Materials and methods: “For multiband peaky speech, the component wave peak latency changes across frequency band were assessed with power law regression (Harte et al., 2009; Neely et al., 1988; Rasetshwane et al., 2013; Strelcyk et al., 2009), conducted in the log-log domain with linear mixed effects regression using the lme4 and lmerTest packages in R (Bates et al., 2015; Kuznetsova et al., 2017; R Core Team, 2020). The parameter *a* of the power law regression was estimated by adding an assumed synaptic delay of 0.8 (Eggermont, 1979; Strelcyk et al., 2009) to the I-V inter-wave delay from the subjects' responses to broadband peaky speech. For subjects who did not have an identifiable wave I in the broadband peaky response, the group mean I-V delay was used – this occurred for 1 of 22 subjects in experiment 1, and 2 of 11 subjects for responses to the female narrator in experiment 2. Because only multiband peaky speech was presented in experiment 3, the mean I-V intervals from experiment 2 were used for each subject for experiment 3. Random effects of subject and each frequency band term were included to account for individual variability that is not generalizable to the fixed effects.”

– The fact that differences between narrators leads to changes to the ABR response is to be expected, and was already reported in Maddox and Lee, 2018. We are not sure why this issue needs to be examined and discussed at such length here. The space devoted to discussing the recording time also seems very long. Neither Abstract or Introduction refers to these topics, and they seem to be side-issues that could be summarised and discussed much more briefly.

It’s true that there were subtle differences between narrators in the 2018 paper, but the differences here are larger and likely stem from differences in mean F0, which is not necessarily the case for the previous paper. As a methods paper we feel it is important to quantify how choice of narrator impacts the response as well as the recording time (and thus how many conditions may be feasible in an experiment). Recording time and the effect of narrator choice are important practical considerations for using this new technique. We added this to the Abstract: “We further demonstrate the versatility of peaky speech by simultaneously measuring bilateral and ear-specific responses across different frequency bands, and discuss important practical considerations such as talker choice.” We also cropped some text in the Results and Discussion sections.

– The authors motivate the work from the use of naturalistic speech, and the application of the developed method to investigate, for instance, speech-in-noise deficits. But they do not discuss how comprehensible the peaky speech in fact is. We would therefore like to see behavioural experiments that quantitatively compare speech-in-noise comprehension, for example SRTs, for the unaltered speech and the peaky speech. Without such a quantification, it is impossible to fully judge the usefulness of the reported method for further research and clinical applications.

The reason we did not do any experiments quantifying the intelligibility of the speech is because it sounds pretty much like normal speech and the spectrograms (as shown in Figure 13 in the Materials and methods section) are essentially identical, so it never occurred to us to test it in this methods paper. There were some example stimulus files attached as supplementary data with reference to these files in the manuscript, but we could have done a better job pointing out how intelligible the speech is in the manuscript, and we have done so in our revisions:

In the Discussion we say “Regardless, our peaky speech generated robust canonical responses with good SNR while maintaining a natural-sounding, if very slightly “buzzy,” quality to the speech.”

We added the following to the Introduction: “The design goal of peaky speech is to re-synthesize natural speech so that its defining spectrotemporal content is unaltered – maintaining the speech as intelligible and identifiable – but its pressure waveform consists of maximally sharp peaks so that it drives the ABR as effectively as possible (giving a very slight “buzzy” quality when listening under good headphones; Audio files 1–6).”

The comment says that it's impossible to judge the usefulness, but we feel that this behavioral experiment testing the intelligibility in noise is best left for a follow-up study with such a focus. Addressing it experimentally now, especially with major slowdowns related to a worsening coronavirus situation, would add a lengthy delay to publication.

We asked the editors for their advice before we submitted this revision that does not include the behavioral data. They indicated that they would be happy for us to submit a revised version of our manuscript on the basis of these revisions including a discussion of the sample files pointing out that the peaky speech is clearly intelligible.

– The neural responses to unaltered speech and to peaky speech are analysed by two different methods. For unaltered speech, the authors use the half-wave rectified waveform as the regressor. For peaky speech, however, the regressor is a series of spikes that are located at the timings of the glottal pulses. Due to this rather different analysis, it is impossible to know to which degree the differences in the neural responses to the two types of speech that the authors report are due to the different speech types, or due to the different analysis techniques. The authors should therefore use the same analysis technique for both types of speech. It might be most sensible to analyse the unaltered speech through a regressor with spikes at the glottal pulses a well. In addition, it would be good to see a comparison, say of a SNR, when the peaky speech is analysed through the half-wave rectified waveform and through the series of spikes. This would also further motivate the usage of the regressor with the series of spikes.

We wondered the same, and these analyses were in the supplemental information where they were easy to miss. Therefore, we moved the analyses and figures to the main body of the paper and added the responses of unaltered speech derived with a pulse train regressor. We kept the figure of the multiband peaky speech vs unaltered speech with the half-wave rectified audio regressor as a supplemental figure (Figure 4—figure supplement 1) since we already have a lot of figures in the main paper. The results show that the new analysis is only possible with the new stimuli. The responses for unaltered and broadband peaky speech are similar when using the half-wave rectified audio as the regressor, and the SNR is similar. The SNR is indirectly shown in the cumulative density functions, which show the time for responses to reach 0 dB SNR. The CDF for using the half-wave rectified audio shows that for ABR lags the rectified regressor takes much longer to reach 0 dB SNR.

The following text from the supplemental figure caption was added to the main body text of the broadband peaky speech results:

“We verified that the EEG data collected in response to broadband peaky speech could be regressed with the half-wave rectified speech to generate a response. […] The aligned phases of the broadband peaky response allow for the distinct waves of the canonical brainstem and middle latency responses to be derived using the pulse train regressor.”

The following text from the supplemental figure caption was added to the main body text of the multiband peaky speech results:

“We also verified that the EEG data collected in response to multiband peaky speech could be regressed with the half-wave rectified speech to generate a response. Figure 4—figure supplement 1 shows that the derived responses to unaltered and multiband peaky speech were similar in morphology when using the half-wave audio as the regressor, although the multiband peaky speech response was broader in the earlier latencies. Correlation coefficients from the 22 subjects for the 0–40 ms lags had a median (interquartile range) of 0.83 (0.77–0.88), which were slightly smaller than the correlation coefficients obtained by re-analyzing the data split into even/odd epochs that each contained an equal number of epochs with EEG to unaltered speech and peaky broadband speech (0.89, interquartile range 0.83–0.94; Wilcoxon signed-rank test W(21) = 58.0, p = 0.025). This means that the same EEG collected to multiband peaky speech can be flexibly used to generate the frequency-specific brainstem responses to the pulse train, as well as the broader response to the half-wave rectified speech.”

– Subsection “Frequency-specific responses show frequency-specific lags”. What causes the difference between the effect of high-pass filtering and subtracting the common response? If they serve the same purpose, but have different results, this raises the question which is more appropriate.

Thank you for this comment—we can see how people following our methods could have been confused or misguided. The way to remove the common component is to calculate it and subtract it. At least for the narrators we tested, the common component contains minimal energy above 150 Hz, so if you’re high-passing at 150 Hz anyway for the ABR, then the step of formally subtracting the common component may not be necessary. We revised the section, which now reads:

“This coherence was due to all pulse trains beginning and ending together at the onset and offset of voiced segments and was the source of the low-frequency common component of each band’s response. The way to remove the common component is to calculate the common activity across the frequency band responses and subtract this waveform from each of the frequency band responses …. Of course, the subtracted waveforms could also then be high-pass filtered at 150 Hz to highlight earlier waves of the brainstem responses, as shown by the dashed lines in Figure 6B. However, this method reduces the amplitude of the responses, which in turn affects response SNR and detectability. In some scenarios, due to the low-pass nature of the common component, high-passing the waveforms at a high enough frequency may obviate the need to formally subtract the common component. For example, at least for the narrators used in these experiments, the common component contained minimal energy above 150 Hz, so if the waveforms are already high-passed at 150 Hz to focus on the early waves of the ABR, then the step of formally subtracting the common component may not be necessary. But beyond the computational cost, there is no reason not to subtract the common component, and doing so allows lower filter cut-offs to be used.”

– Subsection “Frequency-specific responses show frequency-specific lags” paragraph four. This seems a misinterpretation. The similarity between broadband and summated multiband responses indicates that the band filtered components in the multiband stimulus elicited responses that add linearly in the broadband response. It does not imply that the responses to the different bands originate from non-overlapping cochlear frequency regions.

We agree that it shows that they represent a “whole” response separated into parts. They would not add linearly if certain parts of the cochlea were 100% active for two different ones. We moderated this claim as follows:

“The similarity verifies that the frequency-dependent responses are complementary to each other and to the common component, such that these components add linearly into a “whole” broadband response. If there were significant overlap in the cochlear regions, for example, the summed response would not resemble the broadband response to such a degree, and would instead be larger.”

– Is this measure of SNR appropriate, when the baseline is artificially constructed by deconvolution and filtering? Perhaps noise level could be assessed by applying the deconvolution to a silent recording instead? It might also be useful to have a measure of the replicability of the response.

SNR estimate: Looking at the prestimulus baseline is akin to looking at the recording during the wrong stimulus, but with the advantage of having the same EEG state that the subject was in at the time of that recording (and thus the same amount of noise for that epoch) rather than at some other time in the recording session.

To confirm this we computed the wrong-stimulus responses (i.e, we shifted the stimulus trial by 1 so that we used the same EEG but the wrong stimulus trial for the regressor) for every subject and condition. We correlated the SNR estimates based on these shifted responses (using both the pre-stimulus baseline time windows and the response time windows) to the SNR estimates based on the pre-stimulus baseline of the real responses, which are shown in Author response image 1. The correlations indicate that both methods provide similar estimates of SNR, but we prefer our method because it applies to a broader range of scenarios. If the reviewer feels like we should add this figure to the paper we can but out of respect for how lengthy the paper already is, we have left it out for now.

**Author response image 1. respfig1:** SNR (dB) calculations based on the response pre-stimulus baseline as used in the paper versus based off the pre-stimulus baseline and stimulus response window of the responses calculated to the wrong stimulus. The same EEG was used for each calculation. SNRs groups around the unity line, except for similar numbers of responses that were better/worse than the paper’s SNR calculator for those responses that have poor SNR (<-5 dB SNR).

Replication: We have included in Author response image 2 a figure of replicated ABRs of the broadband peaky speech high-pass filtered at 150 Hz. The correlation coefficients are very high. We added added text to first Results section at the end of first paragraph:

“Responses containing an equal number of epochs form the first and second half of the recording had a median (interquartile range) correlation coefficient of 0.96 (0.95 – 0.98) for the 0 – 15 ms lags, indicating good replication.”

But for other sections in the paper we already report several correlations of even/odd splits, which are one kind of measure of replicability. We worry that adding even more analyses or figures could bog down the paper.

**Author response image 2. respfig2:** Replication of ABR responses to broadband peaky speech for the 22 subjects in experiment 1. Equal numbers of epochs for the first (black line) and second (gray line) halves of the recording were included in each replication. Correlation coefficients are provided in the top right corner for individual subjects. The median and interquartile range (IQR) are provided in the top right corner for the grand mean responses in the bottom right subplot.

– "wave III was clearly identifiable in 16 of the 22 subjects": Figure 1 indicates that the word "clearly" may be somewhat generous. It would be worthwhile to discuss wave III and its identifiability in more detail (perhaps its identifiability/non-universality could be compared with that of another less prominent peak in traditionally obtained ABRs?).

An addition to our analyses, which adjusts our analysis for the small clock differences between the sound card and EEG equipment, has improved the early wave components of the responses. Now 19 of 22 subjects had responses with visible wave III. But we take the point that we should state rather than opine. We tempered our description of how clearly identifiable was the wave III. To get a sense for how these rates compare, we went back to the click-evoked responses from Maddox and Lee (2018), and 16/24 showed an identifiable wave III. Morphology was similar. We updated the text to read:

“Waves I and V were identifiable in responses from all subjects (N = 22), and wave III was identifiable in 19 of the 22 subjects. The numbers of subjects with identifiable waves I and III in these peaky speech responses were similar to the 24 and 16 out of 24 subjects for the click-evoked responses in Maddox and Lee (2018).”

[Editors' note: further revisions were suggested prior to acceptance, as described below.]

This manuscript presents an innovative alteration to speech to make it more peaky, resulting in stronger and better delineated responses in the auditory brainstem. Recent work has indeed employed naturalistic speech to investigate subcortical mechanisms of speech processing. However, previous methods were ill equipped to tease apart the neural responses in different parts of the brainstem. The authors show that their speech manipulation improves this issue: the peaky speech that they develop allows them to segregate different waves of the brainstem response. This development allows further and more refined investigations of the contribution of different parts of the brainstem to speech processing, as well as to hearing deficits.The authors have made substantial additions and changes in the revised manuscript to complete and improve the analysis. These are overall satisfactory, but we still have some remaining points that we would like the authors to address.– Due to the additions, the Results section is now even longer, as none of the original text has been removed. We suggest to shorten it by moving some technical details from the Results section to the Materials and methods. For instance, this could concern the details of the peak picking, recording time, waveform repeatability (SNR analysis is implicit in the Recording time section) and details of statistical models. Furthermore, some sentences in the Results seem to serve only to summarize the previous section, which seems unnecessary, or make a general statement that is more suitable for the Discussion.

We have made several cuts to the Results section, making it more than 1200 words shorter than the previous version. Some information (e.g., technical details) has been moved to Materials and methods, some has been condensed into tables (parameters from latency fits), and many other details have been deemed extraneous and removed (e.g., specific details of ICC fits, when all were ≥0.9).

– "unless presented monaurally by headphones". So these studies did show detectable wave I/III waves when using monaural headphone presentation? This should be more clearly phrased. It seems strange to make this presentation mode sound like a limitation – monaural presentation is standard in clinical audiology. This point raises further questions that had not previously occurred to us: can you clarify whether the peaky speech has also been tested using monaural presentation, and if so how the responses differ from diotic/dichotic presentation (and if not, why was diotic presentation chosen)? How would you confirm that dichotic presentation of the peaky speech produces purely ear-specific responses? Could there be potential binaural interactions or contralateral suppression mechanisms with diotic or dichotic presentation? These questions are relevant and should be briefly addressed.

We have changed this to say that no waves before wave V were reported. The original text was based on a figure from Miller’s 2017 patent showing a single subject’s response with apparent earlier waves, but their subsequent 2019 paper shows no hint of waves I or III. We are taking the peer-reviewed paper as the source of record, and have removed the patent citation.

We did diotic recordings because it makes responses bigger. We did dichotic recordings when we wanted separate responses from each ear, and we now note that “It is also possible that there was some contralateral suppression in our dichotic recordings, however it is unlikely that separate monaural presentation would enlarge responses enough to be worth the doubled recording time.”

The dichotic responses are guaranteed mathematically to be ear-specific because the regressors are orthogonal and correspond to stimuli which were presented to one ear and not the other (much in the same way the sham regressors in Figure 15—figure supplement 2 produce zero responses.)

– We are not sure what "region-specific" means here – please clarify.

We revised the text to: “…a speech-evoked response with the same components would assess subcortical structure-specific encoding…”

– It is more common and reliable to use different peak pickers, rather than the same twice. It probably won't matter much here, but it would be preferable in future analyses.

Agreed, we will do this in future analyses.

– It should be explained why the comparison is shown only for the high-pass filter at 30 Hz. The high-pass filter at 150 Hz seems more interesting, as it shows the wave I and III peaks in the peaky speech response that the unaltered speech does not.

This is a good point. We now explain that “…(unlike the broadband peaky response, the 150 Hz high-pass cutoff does not reveal earlier components in the response to unaltered speech).”

– "completed two models". It might be more accurate to write "performed two simulations".

The revision was made: “To better understand the source of this pre-stimulus component – and to determine whether later components were influencing the earliest components – we performed two simulations:.…”

– Subsection “More components of the ABR and MLR are present with broadband peaky than unaltered speech” paragraph three: Does "trial" here mean the time period starting at a glottal pulse? How long did this time period extend from there? From what time point on was the ABR zero-padded? These details should be clarified. It could be informative to also show the input kernels in Figure 3, i.e. the simulated "ABR" that did not show the pre-stimulus component.

We have clarified these details and included the ABR kernel for the linear simulation as a supplemental figure. “… we performed two simulations: (1) a simple linear deconvolution model in which EEG for each 64 s epoch was simulated by convolving the rectified broadband peaky speech audio with an ABR kernel that did not show the pre-stimulus component (Figure 3—figure supplement 1): the average broadband peaky speech ABR from 0 to 16 ms was zero-padded from 0 ms to the beginning of wave I (1.6 ms), windowed with a Hann function, normalized, and then zero-padded from −16 ms to 0 ms to center the kernel;…”

– Cortical responses are much larger and likely to be more affected by attention than the much smaller earliest responses. Is it known whether a high-pass filter at 150 Hz is sufficient to reliably remove all cortical contributions within the amplitude range of the ABR wave I? Can a reference be provided for this?

The cortical contribution to responses in this frequency range is debated (by, e.g., Coffey, Bidelman, and others). However, our modeling – which only includes brainstem components –strongly suggests that the broader pre-stimulus component can result from an imperfect fit of brainstem responses alone, and that the filtering can effectively remove it.

– Strictly speaking, the stimuli differed not only in the narrator's identity but also in text. It would be interesting to comment on whether different texts with the same narrator are likely to give different responses.

We now note in the text that “These high odd-even correlations also show that responses from a single narrator are similar even if the text spoken is different. The overall male-female narrator differences for the ABR indicate that the choice of narrator for using peaky speech impacts the morphology of the early response.”

– Figure 3 and text related to the more aggressive high-pass filtering at 200 Hz:With this more aggressive filter, the 8 ms peak has flipped sign and gained additional latency. This needs to be explicitly acknowledged and addressed. It might be due to the peak of that filter's non-linear group delay at its cutoff frequency, combined with an abundance of signal power at the cutoff frequency. But there could be other reasons as well.

That large negative peak, while it draws the eye, is actually a common result of the more aggressive high-pass filtering. Wave V is still positive going and in fact has a shorter latency (another expected effect). We added this to the text: “As expected from more aggressive high-pass filtering, wave V became smaller, sharper, and earlier, and was followed by a significant negative deflection.”

– A number of parameter estimates from the power law regression results are written as dimensionless but actually have units of ms. This includes τ_synaptic_ (should be 0.8 ms in several locations), τ_I-V_ (in several locations), and a (also in several locations).

Thank you for catching this. For brevity’s sake, these parameters have been moved to a table and their units are now specified.